# Bioengineered bacteria-derived outer membrane vesicles as a versatile antigen display platform for tumor vaccination via Plug-and-Display technology

Keman Cheng[1,2,3,5], Ruifang Zhao[1,2,5], Yao Li[1,2,3], Yingqiu Qi[1,2], Yazhou Wang[1,2], Yinlong Zhang [1,2], Hao Qin[1,2], Yuting Qin[1,2], Long Chen[1,2], Chen Li[1,2], Jie Liang[1,2], Yujing Li[1,2], Jiaqi Xu[1,2], Xuexiang Han[1,2], Gregory J. Anderson[4], Jian Shi[1,2], Lei Ren[3], Xiao Zhao[1,2 ✉] & Guangjun Nie [1,2 ✉]

An effective tumor vaccine vector that can rapidly display neoantigens is urgently needed. Outer membrane vesicles (OMVs) can strongly activate the innate immune system and are qualified as immunoadjuvants. Here, we describe a versatile OMV-based vaccine platform to elicit a specific anti-tumor immune response via specifically presenting antigens onto OMV surface. We first display tumor antigens on the OMVs surface by fusing with ClyA protein, and then simplify the antigen display process by employing a Plug-and-Display system comprising the tag/catcher protein pairs. OMVs decorated with different protein catchers can simultaneously display multiple, distinct tumor antigens to elicit a synergistic antitumour immune response. In addition, the bioengineered OMVs loaded with different tumor antigens can abrogate lung melanoma metastasis and inhibit subcutaneous colorectal cancer growth. The ability of the bioengineered OMV-based platform to rapidly and simultaneously display antigens may facilitate the development of these agents for personalized tumour vaccines.

[1] CAS Key Laboratory for Biomedical Effects of Nanomaterials and Nanosafety & CAS Center for Excellence in Nanoscience, National Center for Nanoscience and Technology of China, Zhongguancun, Beijing, China. [2] Center of Materials Science and Optoelectronics Engineering, University of Chinese Academy of Sciences, Beijing, China. [3] Department of Biomaterials, Key Laboratory of Biomedical Engineering of Fujian Province, College of Materials, Xiamen University, Xiamen, Fujian, China. [4] Iron Metabolism Laboratory, QIMR Berghofer Medical Research Institute, Brisbane, QLD, Australia. [5] These authors contributed equally: Keman Cheng, Ruifang Zhao. ✉email: zhaox@nanoctr.cn; niegj@nanoctr.cn

mmunotherapy is becoming a highly effective cancer treatment and will soon join surgery, chemotherapy, and radiotherapy as a mainstay of cancer therapy[1–3]. The profound somatic mutations in cancer cells lead to altered protein sequences that often generate tumor neoantigens, which are absent from normal tissues and can be presented as neoepitopes on major histocompatibility complex (MHC) molecules[4,5]. Preclinical studies have shown that the recognition of tumor neoantigens by the immune system is a key event in the success of immunotherapy in oncology[6]. However, only about 1% of these "foreign" neoantigens in cancer cells are spontaneously presented to the immune system[7]. Therefore, using efficient vaccine vectors to display and present tumor antigens is a major strategy in the development of effective anticancer therapeutics[8,9], as demonstrated in several clinical trials[10–12].

Outer membrane vesicles (OMVs) are natural, non-replicative particles, with a size of 30–250 nm, secreted by Gram-negative bacteria[13,14]. These vesicles are a crucial element of bacterial homeostasis and an important facilitator of bacteria communication[15,16]. Due to their particulate nature and innate composition, containing pathogen-associated molecular patterns (PAMPs), OMVs possess excellent intrinsic immunostimulatory properties and can act as pathogen mimetic adjuvants[17,18]. Although the immunogenicity of OMVs limits their use as natural nanoparticles for drug delivery, these same immunomodulatory characteristics, the capacity of the vesicles to accumulate in lymph nodes due to their small size and natural composition, and their ability to be manufactured in large quantities by bacterial fermentation, make OMVs attractive candidates as vaccine vectors. This potential has already been realized to generate OMV-based vaccines against pathogenic microorganisms[19]. OMVs collected directly from bacterial pathogens can induce a strong specific immune response against the given pathogen. For example, the OMV-based Group B meningococcal vaccine MeNZB has effectively limited the incidence and mortality of meningitis in New Zealand[20]. More importantly, via genetic engineering, homologous or heterologous antigens from different pathogens or other sources can be embedded in OMVs[21]. For example, E. coli-derived OMVs expressing antigens from influenza virus, Plasmodium, Pneumococcus, Chlamydia, or Group A and B Streptococcus can stimulate the body to produce specific antibodies against these pathogenic microorganisms[22–26]. These studies have demonstrated the great potential of OMVs as a vaccine platform that can stimulate humoral immunity. OMVs may also have the potential to stimulate cellular immunity for cancer therapy, but there has been little progress in this endeavor at this time.

Exploiting bacterial components as tumor immunotherapeutic agents is not a new idea. It was first performed in the early 1890s by William Coley in what was the beginning of the age of immuno-oncology[27]. Coley developed a cocktail of weakened bacteria for cancer treatment; this solution was known as Coley's toxins[27]. Compared to the weakened bacteria, OMVs are safer because the preparation is acellular. In 2017, Kim et al. found that OMVs from E. coli could effectively induce interferon-γ (IFNγ)-mediated long-term anti-tumor immune responses[28], however, their study did not utilize OMVs decorated with tumor antigens to stimulate antigen-specific anti-tumor immunity, as the technology to accomplish this had not yet been developed. Tumor antigens are diverse and can vary considerably between patients, making it impractical to produce a single antigen-decorated OMV-based tumor vaccine that is effective for all patients[29]. Constructing a functional OMVs platform that can rapidly display different tumor antigens and simultaneously display multiple tumor antigens are the keys to the development of personalized tumor vaccines.

In this work, we establish a flexible tumor vaccine platform based on OMVs, using genetic engineering and "Plug-and-Display" technology[30–32] to display the target antigens. We first show that tumor antigens can be displayed on the surface of OMVs as ClyA fusion proteins. Secondly, we verify that the tumor antigens displayed by the ClyA protein can induce T-cell-mediated, specific anti-tumor immunity. Finally, we employ the protein Plug-and-Display system, including a SpyTag (SpT)/SpyCatcher (SpC) pair[31] and a SnoopTag (SnT)/SnoopCatcher (SnC) pair[32], in which the protein tag can spontaneously bind to the protein catcher through isopeptide bond formation. By expressing the protein catchers as fusion proteins with ClyA, various tumor antigens linked to protein tags can be rapidly and simultaneously displayed on the OMVs surface. The OMVs efficiently accumulate in draining lymph nodes by virtue of their small size (the "nano-size effect") and biomimetic "foreigner" status, and it is here that the OMVs are processed and the tumor antigens are presented by dendritic cells (DCs). This bioengineered OMV-based vaccine platform enables flexible tumor antigen display and specific anti-tumor immunity in preclinical cancer models.

## Results

**Engineered display of heterologous proteins or antigen peptides on the OMVs surface.** The display of heterologous proteins on OMVs was achieved by fusing the proteins to the ClyA protein using recombinant DNA technology. ClyA is one of the most abundant proteins on the OMVs surface[33]. To optimize conditions for the expression of the fusion proteins, the genes encoding ClyA and the indicator enzyme luciferase (ClyA-Luc, CL) were recombined in the plasmid pET28a, and the fusion protein was expressed in E. coli Rosetta (DE3). Some of the key variables affecting the expression were examined (Supplementary Fig. 1), and the optimal expression conditions were determined to be 16 °C for 14 h under 0.1 mM isopropyl-β-d-thiogalactoside (IPTG) induction with shaking at 160 rpm. As shown in Fig. 1a, luciferase (Luc) expression was not detected in the ClyA-none (CN) group (only ClyA expression without any fusion). While considerable expression of Luc by the bacteria was observed when the Luc gene was expressed on its own, its incorporation into OMVs was only observed when it was expressed as a fusion protein with ClyA, indicating an efficient expression of CL in OMVs (Fig. 1a). After adding the luciferase substrate (fluorescein potassium), fluorescence was observed in the CL group but not in the CN group (Fig. 1b), indicating that the expressed heterologous protein retained its biological activity. This suggests that the expressed protein retained its biological structure, which is an important characteristic for successful antigen presentation. We then changed the displayed protein Luc to an antigenic epitope of ovalbumin (OVA), $OVA_{257-264}$ (SIINFEKL), and three HA tags. Expression of the fusion protein of ClyA and $OVA_{257-264}$ (ClyA-OVA, CO) within OMVs was verified by western blot analysis by probing the HA tags (Fig. 1c). The morphology and sizes of CN, Luc, CL, and CO OMVs were characterized by transmission electron microscopy (TEM) and dynamic light scattering (DLS; Fig. 1d and Supplementary Fig. 2a–c). All OMVs showed a bilayer structure and a uniform circular morphology with a diameter of ~30 nm.

**Innate immune response and antigen-specific T-cell-mediated anti-tumor immunity induced by OMVs displaying tumor antigen peptide.** We next tested whether the OMVs displaying tumor antigen peptide could stimulate an effective immune response by activating bone marrow-derived dendritic cells (BMDCs). We first confirmed that OMVs were not toxic to murine BMDCs in the concentration range used in this study,

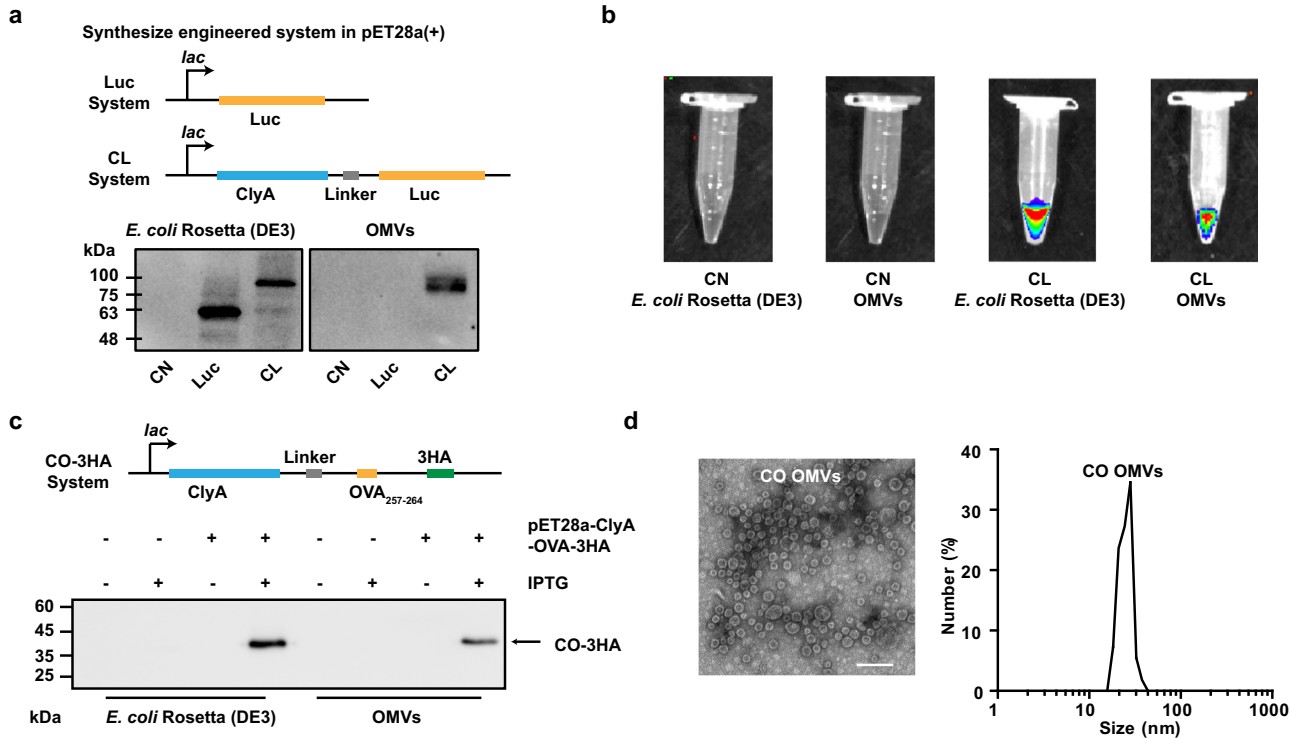

**Fig. 1 Engineered display of heterologous proteins or antigen peptides on the OMVs surface. a** Schematic representation of the pET28a-Luc and pET28a-ClyA-Luc construct, and western blot analysis of ClyA-none (CN), Luc, and ClyA-Luc (CL) expression in *E. coli* Rosetta (DE3), and OMVs derived from the bacteria. **b** In vitro bioluminescence images to demonstrate luciferase enzyme activity following expression in Rosetta (DE3) and OMVs. **c** Schematic representation of the pET28a-ClyA-OVA-3HA construct, and western blot analysis of 3× HA-tagged ClyA-OVA (ClyA-OVA-3HA, CO-3HA) expression in *E. coli* Rosetta (DE3) and OMVs. pET28a-ClyA-OVA-3HA was the expression plasmid, and IPTG was the expression inducer. **d** TEM image and DLS analysis of ClyA-OVA OMVs (CO OMVs). Scale bar, 100 nm. Source data are provided as a Source Data file.

using annexin V-APC/7-AAD apoptosis detection assay to stain dead cells (Supplementary Fig. 3). After engulfing antigens and adjuvants, antigen-presenting cells normally respond in two main ways to trigger a subsequent T-cell-mediated adaptive immune response[34,35]. First, the antigens bind to major histocompatibility complex (MHC) I molecules (MHCI antigen) for presentation to T lymphocytes. Second, the adjuvants drive the activation of the immature DCs. The resulting mature DCs express the co-stimulatory molecules CD80 and CD86 on their plasma membrane, and these act in synergy with the MHCI antigen to enhance the T-cell response. Thus, we investigated the effect of OMVs expressing chimeric proteins on antigen presentation and maturation in DCs. Since OMVs are a natural immune adjuvant, we observed a significant increase in the proportion of CD80+ and CD86+ cells after CD11c+ BMDCs were cultured with CN OMVs, OVA$_{257-264}$ + CN OMVs or CO OMVs (Fig. 2a, b and Supplementary Fig. 4). Importantly, the presentation of OVA$_{257-264}$ by MHCI H-2Kb (MHCI-OVA) was detected in BMDCs and DC2.4 in the OVA$_{257-264}$ and CO OMVs groups (Fig. 2c and Supplementary Fig. 5a, b), while there was almost no MHCI-OVA detected in Pan 02 tumor cells (Supplementary Fig. 5c), indicating that OVA$_{257-264}$ is only effectively presented by DCs (DC2.4 and BMDCs). The immune response stimulated by DC maturation was further studied by measuring the secretion of pro-inflammatory cytokines into the culture medium. Compared to the PBS and OVA$_{257-264}$ groups, the secretion of TNF-α, IL-6, and IL-1β increased significantly after treatment of BMDCs with any of the OMVs groups for 3 h (Fig. 2d–f). In summary, OMVs expressing the chimeric protein effectively induced DC maturation and antigen presentation. The ability to stimulate both the innate and adaptive immune systems in a single physical unit is notable, as the mixing of the separate components is far less effective than the co-delivered formulation (Fig. 2c and Supplementary Fig. 5a, b).

Next, we evaluated OVA$_{257-264}$-specific immunity and its anti-tumor effects in vivo. Mice were challenged by tail vein injection of OVA-expressing B16 cells (B16-OVA) and immunized with saline, OVA$_{257-264}$, CN OMVs, OVA$_{257-264}$ + CN OMVs or CO OMVs subcutaneously on days 3, 6, and 11 after B16-OVA administration (Fig. 2g). On day 17, we collected the inguinal lymph nodes and found these to be significantly enlarged in the groups immunized with CN OMVs, OVA$_{257-264}$ + CN OMVs or CO OMVs (Supplementary Fig. 6a). The percentages of CD80+ and CD86+ DCs within those lymph nodes also dramatically increased in these groups relative to those treated with saline or OVA$_{257-264}$, indicating that OMVs could significantly activate the innate immune system and promote DC maturation in vivo (Fig. 2h, i). However, activation of innate immunity alone (CN OMVs group) reduced the number of B16-OVA lung metastatic nodules only to a limited extent (Fig. 2j, k), which is consistent with the previous report[28]. OVA$_{257-264}$ + CN OMVs immunization provided a significantly enhanced anti-tumor response, but CO OMVs immunization proved the most effective and almost eliminated B16-OVA lung metastases (Fig. 2j, k). We also assessed the abundance of antigen-specific T lymphocytes in the spleen of immunized mice by flow cytometry and enzyme-linked immunospot (ELISPOT) assay. The proportion of splenocytes rechallenged with OVA$_{257-264}$ antigen made up by IFNγ+ cytotoxic T lymphocytes was significantly higher when the splenocytes were derived from mice in the CO OMVs group, compared to all other groups (Fig. 2l and Supplementary Fig. 6b). Similarly, more IFNγ was produced by the splenocytes from mice

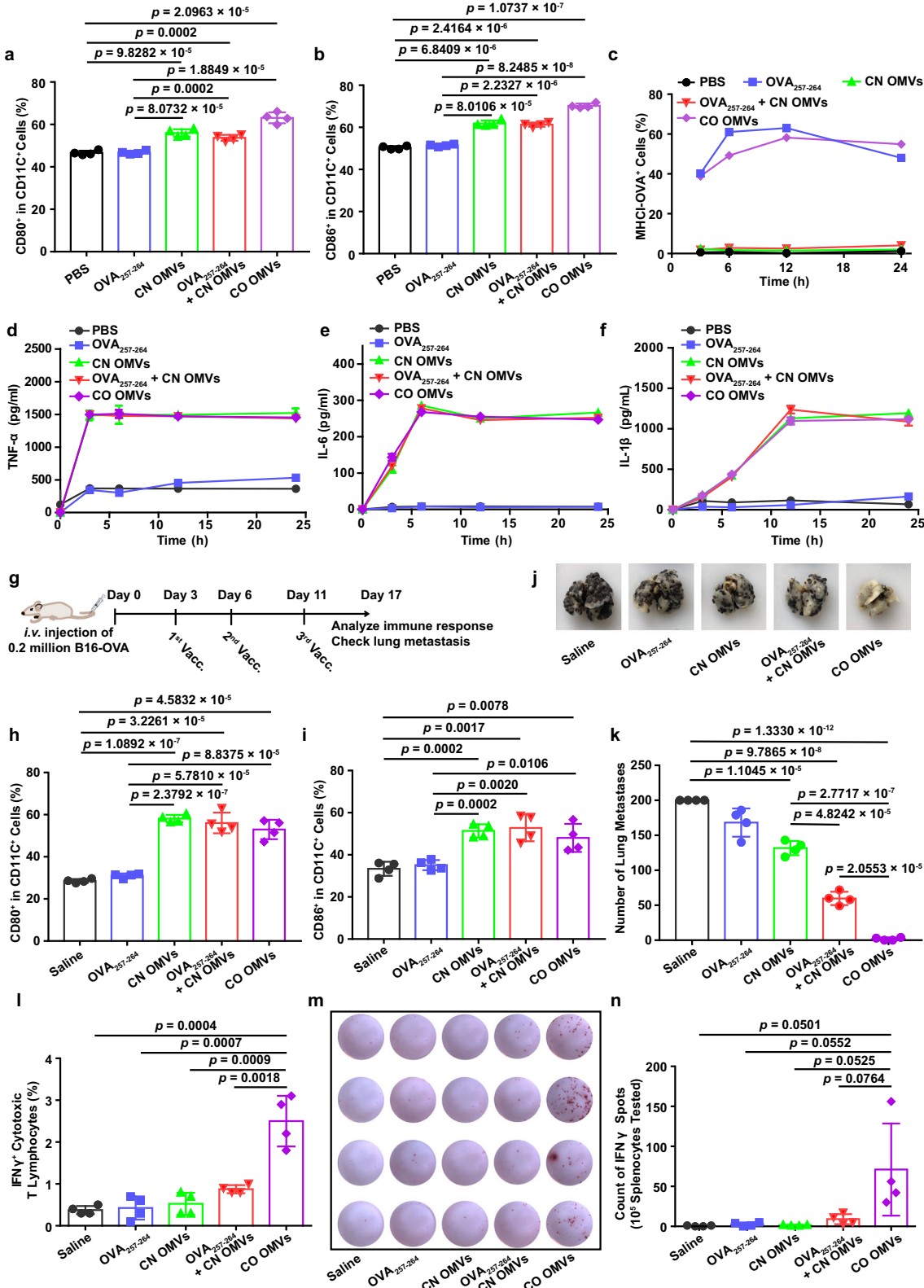

immunized with CO OMVs after exposing the cells to $OVA_{257-264}$, compared with the other groups (Fig. 2m, n). We also analyzed the percentage of T-lymphocyte subpopulations in the inguinal lymph nodes (Supplementary Fig. 7a–e) and blood (Supplementary Fig. 8a–e) from the immunized mice. Immunization with $OVA_{257-264}$ + CN OMVs or CO OMVs significantly increased the numbers of $CD3^+$, $CD3^+CD8^+$, and $CD3^+CD4^+$ T

lymphocytes in the inguinal lymph nodes (Supplementary Fig. 7a–e), however only the mice immunized with CO OMVs showed this effect in the blood (Supplementary Fig. 8a–e), indicating the importance of the linkage between the antigen and OMVs. The levels of lung-infiltrating $CD8^+$ cytotoxic T lymphocytes were also highest in the CO OMVs group, compared with the other groups (Supplementary Fig. 9). Together, these

**Fig. 2 Innate immune response and antigen-specific T-cell-mediated anti-tumor immunity induced by tumor antigen peptide-displayed OMVs.**
**a**, **b** Maturation of BMDCs following treatment with OMVs preparations or controls. Flow cytometry was used to measure the percentage of CD80$^+$ (**a**) or CD86$^+$ (**b**) cells in CD11c$^+$ BMDCs ($n = 4$). **c** The expression of the MHCI-OVA complex on the surface of BMDCs was measured by flow cytometry ($n = 3$). **d**–**f** TNF-α (**d**), IL-6 (**e**), and IL-1β (**f**) levels in the BMDC-conditioned medium after the indicated treatments ($n = 3$). **g** Schema showing the mouse B16-OVA melanoma model used to study the effects of OMVs vaccination (Vacc.) on lung metastasis. C57BL/6 mice were inoculated with B16-OVA melanoma cells ($n = 4$, $2 \times 10^5$ cells/mouse, i.v.), then immunized with the following vaccines: saline, OVA$_{257-264}$, CN OMVs (ClyA-none OMVs), OVA$_{257-264}$ + CN OMVs or CO OMVs 3, 6, and 11 days later. Lung metastasis and immune responses were analyzed on day 17. **h**, **i** The maturation status of DCs in inguinal lymph nodes on days 17 post immunization. The percentage of CD80$^+$ (**h**) or CD86$^+$ (**i**) cells in CD11c$^+$ cells was assessed by flow cytometry. **j**, **k** Lung metastasis was assessed on days 17 after tumor cell administration and following the indicated vaccine treatments. The lungs were photographed (**j**), and the tumor nodules in the lungs were counted (**k**). **l** Flow cytometry analysis of the percentage of IFNγ$^+$ cytotoxic T lymphocytes (CD3$^+$CD8$^+$IFNγ$^+$ T cells) in splenocytes re-stimulated with OVA$_{257-264}$ antigen. **m**, **n** IFNγ secretion, as measured by the ELISPOT assay, from splenocytes which had been re-stimulated with OVA$_{257-264}$ (**m**). Quantitative analysis of the ELISPOT assay for IFNγ secretion is shown in (**n**). **g**–**n**, $n = 4$. The data (**a**–**f**, **h**, **i**, **k**, **l**, **n**) are shown as mean ± SD. Statistical analysis was performed by a two-tailed unpaired $t$ test. Source data are provided as a Source Data file.

data suggest that OMVs are an ideal vaccine vector for displaying tumor antigens to elicit an anti-tumor immunity and that the ClyA protein is an excellent chimeric partner for antigen display.

**Design and characterization of a rapid and flexible OMV-based antigen display platform.** As noted above, mutations that accumulate during tumor progression can generate numerous tumor neoantigens. With technological advances in genome sequencing, genomics, and cancer immunotherapy, these "foreign" antigens can be predicted and then targeted in the design of tumor vaccines to trigger specific anti-tumor immune responses[4,9]. Although OMVs represent a useful platform for tumor vaccine development through the surface expression of chimeric proteins presenting tumor antigens, expanding this system to cater to the diversity and heterogeneity of tumor antigens remains a significant challenge. To overcome this issue, we have utilized the protein Plug-and-Display system comprising the SpT/SpC pair and SnT/SnC pair (Fig. 3a)[31,32]. The SpC and SnC catchers were expressed as fusion proteins with ClyA (ClyA-Catchers, CC) on the OMVs surface, while the SpT- or SnT-labeled antigens can be displayed rapidly on OMVs by binding to CC through isopeptide bond formation between the tag and catcher[31,32]. The lipopolysaccharides (LPS) content in CC OMV was 49.9 and 204.1 ng/mg OMV protein detected by enzyme-linked immunosorbent assay (ELISA) and Limulus amebocyte lysate (LAL) assay, respectively.

We first characterized CC OMVs by TEM and DLS (Fig. 3b) to find a typical bilayer structure and uniform spherical morphology with a diameter of ~30 nm. After repeated freezing and thawing, the morphology and size of CC OMVs remained unchanged, indicating that CC OMVs can be stored at −80 °C (Supplementary Fig. 10a, b). The morphology and size were also unaffected by incubation in 10% fetal bovine serum (FBS) for 24 h, suggesting that CC OMVs are likely to remain stable in the body long enough for vaccination to be effective (Supplementary Fig. 10c). After conjugating with SpT-OVA, repetitive freezing–thawing CC-SpT-OVA OMVs could also effectively stimulate the maturation of BMDCs (Supplementary Fig. 10d) and promote the presentation of antigens (Supplementary Fig. 10e), with no significant difference compare to the non-freezing–thawing (fresh) preparation. Next, to verify that the expressed catchers on the CC OMVs surface can rapidly link and display antigens, we synthesized HA-tagged SpyTag (SpT-HA) and SnoopTag (SnT-HA). The linkage between SpT-HA and SpC on CC OMVs, or SnT-HA and SnC on CC OMVs, was monitored by western blot analysis using an anti-HA antibody. A concentration-dependent increase in the appearance of the ClyA-SpC-SpT-HA or ClyA-SnC-SnT-HA conjugate (which in each case has a M$_r$ of ~45,000 Da) was observed (Fig. 3c, d). The connection between SpC and SpT or SnC and SnT was stable and was unaffected by storage at different

temperatures or treatment with 10% FBS for 24 h (Supplementary Fig. 11a, b). In addition, there was no change in the morphology or size of the vesicles following the binding of the antigen (Supplementary Fig. 11c). To verify whether SpT- and SnT-labeled antigens can be simultaneously bound to the same OMVs, SpT-Cys (Cysteine) and SnT-Cys were conjugated with 5 and 10 nm gold nanoparticles, respectively, and incubated with CC OMVs expressing both SpC and SnC. TEM revealed that gold nanoparticles of both sizes appeared on the OMVs surface at the same time, indicating that CC OMVs can link with and display multiple antigens simultaneously (Fig. 3e). These results show that the catchers were successfully fusion expressed with the ClyA protein on the CC OMVs surface, and the SpT/SnT-labeled antigens can be rapidly displayed at the ClyA site.

To extend this work to a specific immunogen, SpyTag-labeled OVA$_{257-264}$ (SpT-OVA) was displayed on CC OMVs (CC-SpT-OVA OMVs) via the linkage between SpT and SpC (Supplementary Fig. 12). This conjugation did not affect the ability of the OMVs to stimulate the maturation of DCs, as indicated by up-regulation of the surface expression of CD80 and CD86 in BMDCs (Supplementary Fig. 13a, b). We also investigated antigen presentation in BMDCs. The MHCI-OVA complex was observed when the free antigen (SpT-OVA) or antigen-OMVs (CC-SpT-OVA OMVs) preparations were used, but not with the antigen/OMVs mixture (SpT-OVA + CN OMVs; Supplementary Fig. 13c). These results are similar to those shown in Fig. 2c. When fluorescein PE-anti-MHCI-OVA was used to label the MHCI-OVA complex, fluorescence was only detected in the SpT-OVA and CC-SpT-OVA OMVs groups, and the intensity of the PE fluorescence on the cells increased gradually over time (Fig. 3f and Supplementary Fig. 14a). We also evaluated the efficiency of antigen uptake by BMDCs. SpT-OVA was labeled with Cy5.5 (SpT-OVA-Cy5.5), and cellular uptake was monitored by confocal microscopy (Fig. 3g and Supplementary Fig. 14b). Only BMDCs treated with SpT-OVA-Cy5.5 or CC-SpT-OVA-Cy5.5 OMVs exhibited apparent Cy5.5 fluorescence. These data suggest that our OMV-based antigen display platform can effectively deliver antigens to DCs. As noted above, the immune system is most effectively stimulated when the antigen and OMVs are physically linked. Mixing the antigen and OMVs is far less effective. The reasons for this are not fully understood, but due to the ability of DCs to rapidly take up particulate matter, including OMVs, these cells may be induced to differentiate before sufficient antigen is taken up to induce a robust immune response. The covalent linkage between antigen and OMVs was important and necessary for successful antigen presentation. Due to the relatively high uptake rate of the particulate matter, the mixture of OMVs and antigen resulted in the premature maturation of DCs which severely restricted the effective cell uptake of antigen.

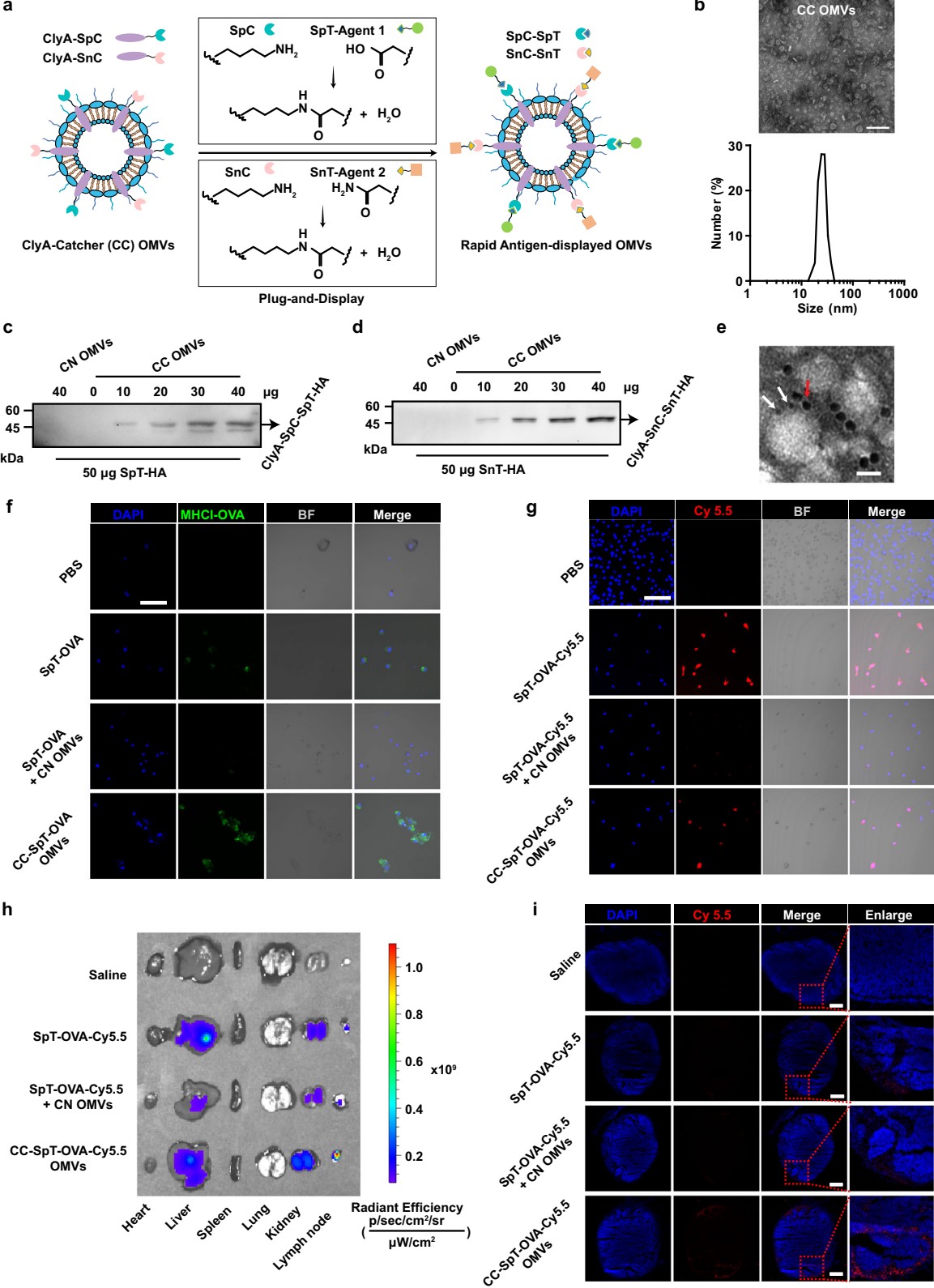

We also examined the ability of the conjugated OMV antigen system to accumulate in draining lymph nodes in vivo and compared this to the situation where the OMVs and antigen were not linked. Saline, SpT-OVA-Cy5.5, SpT-OVA-Cy5.5 + CN OMVs, or CC-SpT-OVA-Cy5.5 OMVs were administered intradermally to mice. Various organs and the inguinal draining lymph nodes of mice were resected and the level of fluorescence was measured at 6 h (Supplementary Fig. 14c) and 12 h (Fig. 3h) post injection. Strong fluorescence was observed in the liver at 6 h, but this declined by 12 h. However, there was a clear accumulation of fluorescence in the lymph nodes at 12 h (Fig. 3h). At this time point, the lymph nodes were sectioned and examined by fluorescence microscopy. Only when the antigen was conjugated to CC OMVs (CC-SpT-OVA-Cy5.5

**Fig. 3 Design and characterization of the flexible OMV-based antigen display platform-antigen presentation on BMDCs and enrichment of antigens and adjuvants in lymph nodes. a** Schematic illustration of ClyA-Catcher (CC) OMVs system for antigen display. SpyCatcher (SpC) and SnoopCatcher (SnC) were expressed as fusion proteins with ClyA (ClyA-Catcher, CC) on the OMVs surface. SpyTag (SpT) or SnoopTag (SnT)-labeled antigens bind to CC OMVs through isopeptide bond formation between the tag and catcher. **b** TEM and DLS analysis of CC OMVs. Scale bar, 100 nm. **c, d** The conjugation of SpT-HA (**c**) or SnT-HA (**d**) to ClyA-Catchers on the CC OMVs surface was verified by western blot analysis using an anti-HA antibody. **e** The simultaneous display of two Catcher/Tag pairs by the same OMVs. SpT-Cys- and SnT-Cys labeled with 5 (white arrow) and 10 nm (red arrow) gold nanoparticles, respectively, were used to identify SpC and SnC on CC OMVs. Scale bar, 20 nm. **f** Confocal microscopy images of antigen presentation by BMDCs incubated with the indicated formulations for 12 h. The cell nuclei were stained blue (DAPI), and MHCI-OVA complexes were stained green (PE-anti-mouse H-2Kb bound to SIINFEKL) ($n = 3$). BF bright field. Scale bar, 50 μm. **g** Confocal microscopy images of antigen uptake by BMDCs after incubation with the indicated OMVs formulations for 12 h. The cell nuclei were stained blue (DAPI), and the antigen was labeled with Cy5.5 (red) ($n = 3$). Scale bar, 50 μm. **h, i** Lymph node accumulation of OMVs in vivo ($n = 3$). Various organs and the inguinal draining lymph nodes of mice were collected 12 h after s.c. immunization with the indicated OMVs formulations to examine the accumulation of Cy5.5 fluorescence (**h**). Frozen sections of the lymph nodes were prepared and examined by fluorescence microscopy (**i**). Cell nuclei were stained blue (DAPI), and the antigen was labeled with Cy5.5 (red). Scale bar, 1 mm. Source data are provided as a Source Data file.

OMVs) was significant fluorescence observed (Fig. 3i). The efficient OMVs accumulation into the lymph nodes is likely attributed to the cell-independent drainage due to their small size and the cell-dependent drainage due to their natural "foreigner" status[19].

**Single-tumor antigen display by catcher-decorated OMVs.** We next investigated the anti-tumor effects of vaccination with CC OMVs displaying one tumor antigen. An antigenic epitope of tyrosinase-related protein 2 (TRP2), TRP2$_{180-188}$ (SVYDFFVWL), was chosen for analysis as it is a well-known tumor-associated antigen in the B16-F10 melanoma model[36,37]. The TRP2$_{180-188}$ was labeled with SnT (SnT-TRP2), then connected to CC OMVs (CC-SnT-TRP2 OMVs) as described above. A mouse B16-F10 melanoma metastasis model was immunized with CC-SnT-TRP2 OMVs, and we compared this to treatment with free SnT-TRP2, CN OMVs, or a mixture of SnT-TRP2 and CN OMVs (Fig. 4a). Vaccination with formulations containing OMVs (including CN OMVs, SnT-TRP2 + CN OMVs, and CC-SnT-TRP2 OMVs) induced the enlargement of the draining lymph nodes (Supplementary Fig. 15a), in which the proportion of CD80$^+$ and CD86$^+$ DCs also increased significantly (Fig. 4b, c). Immunization with CC-SnT-TRP2 OMVs almost eliminated tumor metastasis and induced the strongest infiltration of CD8$^+$ cytotoxic T lymphocytes into the lungs (Fig. 4d, e and Supplementary Fig. 15b). All other formulations were less effective. To evaluate the levels of antigen-specific T lymphocytes, the splenocytes were collected and rechallenged with TRP2$_{180-188}$. Flow cytometry and ELISPOT analyses indicated that CC-SnT-TRP2 OMVs elicited a strong increase in the numbers of IFNγ$^+$ cytotoxic T lymphocytes (Fig. 4f and Supplementary Fig. 16) and IFNγ production (Fig. 4g, h), confirming that OMVs displaying a specific tumor antigen can induce a strong adaptive immune response. The weak triggering of a TRP2$_{180-188}$-specific immune response in the CN OMVs and SnT-TRP2 + CN OMVs groups could be due to the release of TRP2 antigen from tumor cells killed by the OMV-activated innate immune system and the mixed SnT-TRP2, respectively. These results demonstrate that a tumor antigen can be rapidly displayed by bioengineered OMVs, and presented efficiently to T lymphocytes to drive a robust anti-tumor effect.

**Display of two tumor antigens by catcher-decorated OMVs.** The versatility of the OMVs system would be greatly enhanced if the vesicles were able to display multiple antigens. To investigate whether this was feasible in practice, we first tested two antigens from OVA in our OMV-based platform. One of these was OVA$_{257-264}$ (termed "OTI" (an epitope that can stimulate the production of MHC class I-restricted, ovalbumin-specific, CD8$^+$ T cells in mice)). As described above, after a presentation by the MHCI complex, OTI can activate CD8$^+$ T lymphocytes which directly kill cancer cells. The other antigen used was OVA$_{223-339}$ (ISQAVHAAHAEINEAGR; "OTII" (an epitope that can stimulate the production of MHC class II-restricted, ovalbumin-specific, CD4$^+$ T cells in mice)). After a presentation by the MHCII complex, OTII can activate CD4$^+$ T lymphocytes (helper T cells) to enhance the cytotoxicity of CD8$^+$ T cells[38]. OTI and OTII were respectively labeled with SpT (SpT-OTI) and SnT (SnT-OTII). SpT-OTI and SnT-OTII were simultaneously connected to CC OMVs to generate CC-SpT-OTI/SnT-OTII OMVs. A number of control formulations were also prepared, including CC OMVs displaying single peptides (CC-SpT-OTI OMVs or CC-SnT-OTII OMVs) and the mixed formulation (SpT-OTI + SnT-OTII + CN OMVs). In these experiments, the mice were immunized only twice (days 3 and 7 after tumor inoculation) as the efficacy of the formulations was expected to be increased (Supplementary Fig. 17a). As expected, at the end of the treatment, the animals immunized with formulations containing OMVs had larger draining lymph nodes and a higher proportion of CD80$^+$ or CD86$^+$ cells compared to the saline group (Supplementary Fig. 17b–d). Although the OMVs displaying single antigens and the mixed formulation significantly reduced tumor burden, the strongest anti-tumor response was found in mice immunized with CC-SpT-OTI/SnT-OTII OMVs (Fig. 5a, b). Splenocytes from animals in the CC-SpT-OTI/SnT-OTII OMVs group also secreted the most IFNγ when re-stimulated with OVA$_{257-264}$ and OVA$_{223-339}$ (Fig. 5c, d), which is consistent with efficient antigen presentation and the anti-tumor effects. Interestingly, increased proportions of IFNγ$^+$ CD3$^+$CD8$^+$ and CD3$^+$CD4$^+$ cells were found in the mice immunized with the formulations containing OTI and OTII, respectively (Fig. 5e, f). These effects were stronger when the antigens were displayed by the OMVs than when the antigens and OMVs were mixed, but not physically linked. These data demonstrate that the OMV-based platform can simultaneously display different types of antigens to trigger a multiple T-cell-mediated, synthetic anti-tumor immunity.

We next examined the efficacy of the OMVs system displaying two tumor antigens that are both able to elicit CD8$^+$ cytotoxic T-lymphocyte activation. To this end, we simultaneously linked SpT-OTI and SnT-TRP2 to CC OMVs to form CC-SpT-OTI/SnT-TRP2 OMVs. Control formulations included OMVs displaying single peptides (CC-SpT-OTI OMVs or CC-SnT-TRP2 OMVs) and a mixed formulation (SpT-OTI + SnT-TRP2 + CN OMVs). The mice were vaccinated on days 3 and 7 after tumor cell inoculation (Supplementary Fig. 18a). We verified that the innate immune response was activated whenever OMVs were present (Supplementary Fig. 18b–d). The strongest anti-tumor

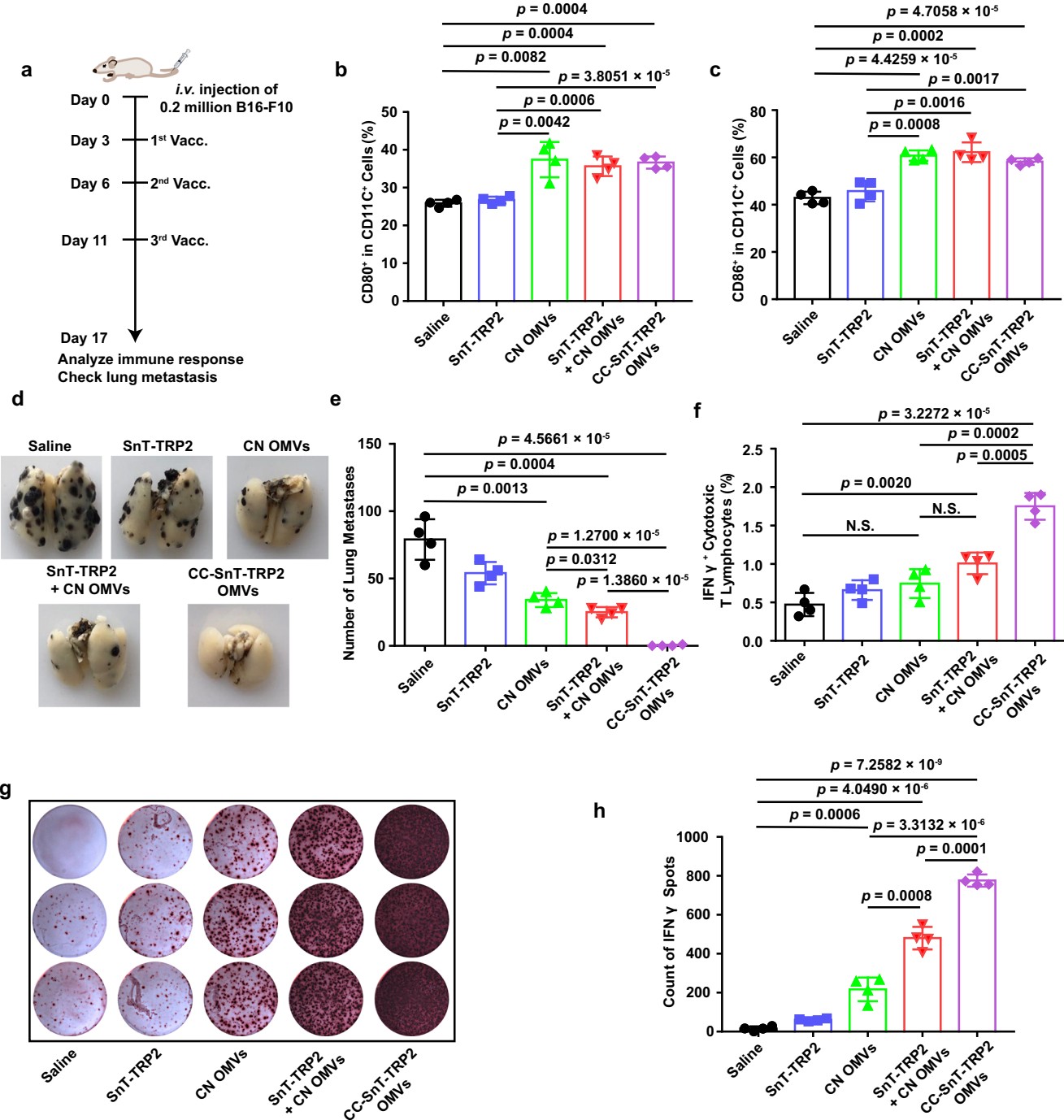

**Fig. 4 Single-tumor antigen (TRP2$_{180-188}$) display by catcher-decorated OMVs. a** Schema showing the B16-F10 melanoma lung metastasis mouse model and the OMVs vaccination (Vacc.) timeline. C57BL/6 mice were inoculated with B16-F10 melanoma cells ($n = 4$, $2 \times 10^5$ cells/mouse, i.v.), then immunized with the following vaccines or controls: saline; SnT-TRP2; CN OMVs; SnT-TRP2 + CN OMVs; or CC-SnT-TRP2 OMVs on days 3, 6, and 11 after inoculation of the tumor cells. Lung metastasis and immune responses were analyzed on day 17. TRP2: tyrosinase-related protein 2. **b**, **c** Analysis of DC maturation in inguinal lymph nodes at the end of the treatment period (day 17). The percentage of CD80$^+$ (**b**) or CD86$^+$ (**c**) cells in CD11c$^+$ cells was determined by flow cytometry. **d**, **e** Lungs were collected at the end of the treatment period and photographed (**d**), then the tumor nodules in the lung were counted (**e**). **f** Flow cytometry analysis of IFNγ$^+$ cytotoxic T lymphocytes in splenocytes re-stimulated with TRP2$_{180-188}$ antigen. The percentage of IFNγ$^+$ cells in the CD3$^+$CD8$^+$ T-cell subpopulation is shown. **g**, **h** IFNγ secretion from splenocytes (as determined by the ELISPOT assay) which had been re-stimulated with TRP2$_{180-188}$ (**g**). Quantitative analysis of the ELISPOT data is shown in (**h**). The data (**b**, **c**, **e**, **f**, **h**) are shown as mean ± SD. Statistical analysis was performed by a two-tailed unpaired $t$ test. N.S. no significance. Source data are provided as a Source Data file.

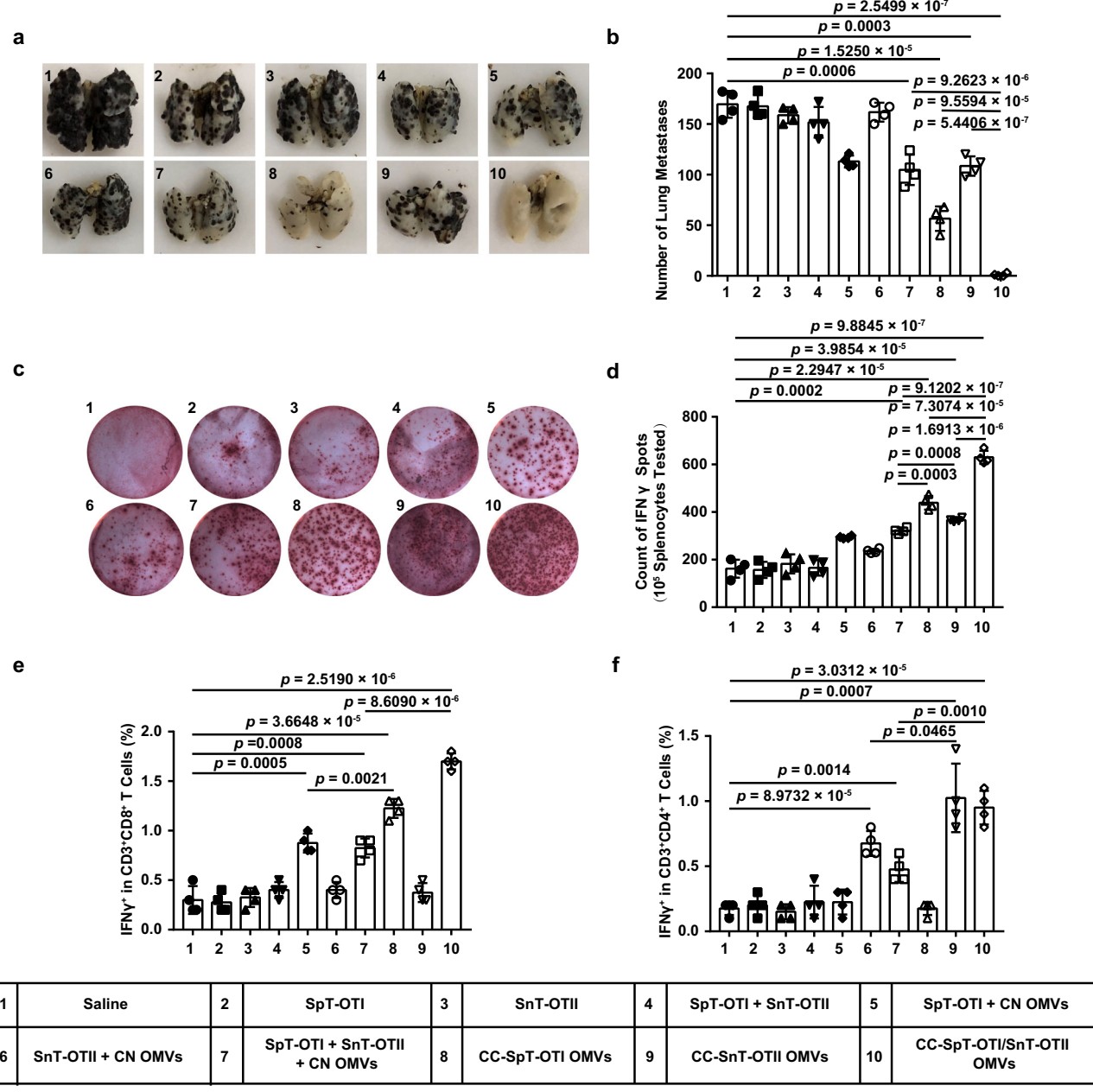

**Fig. 5 Dual-tumor antigen (OVA$_{257-264}$ and OVA$_{223-339}$) display by catcher-decorated OMVs triggers CD4$^+$ and CD8$^+$ T-cell-mediated synthetic anti-tumor immunity.** SpT-OTI and SnT-OTII were displayed either singly (CC-SpT-OTI OMVs or CC-SnT-OTII OMVs) or simultaneously (CC-SpT-OTI/SnT-OTII OMVs) by CC OMVs. Mixed formulations (SpT-OTI + CN OMVs, SnT-OTII + CN OMVs, and SpT-OTI + SnT-OTII + CN OMVs) were used as controls. OTI: OVA$_{257-264}$, OTII: OVA$_{223-339}$. **a**, **b** Lungs were collected on day 17 at the end of the treatment period and photographed (**a**); the tumor nodules were counted (**b**). **c**, **d** IFNγ secretion by splenocytes isolated from the various treatment groups was determined by the ELISPOT assay after re-stimulation of the cells with OVA$_{257-264}$ and OVA$_{223-339}$ (**c**). Quantitative data derived from the ELISPOT assays are shown in (**d**). **e**, **f** Flow cytometry analysis of IFNγ$^+$ cells in the CD3$^+$CD8$^+$ T-cell subpopulation (**e**) or the CD3$^+$CD4$^+$ T-cell subpopulation (**f**) in splenocytes re-stimulated with OVA$_{257-264}$ and OVA$_{223-339}$ antigens. The data (**b**, **d**–**f**) are shown as mean ± SD ($n = 4$). Statistical analysis was performed by a two-tailed unpaired $t$ test. Source data are provided as a Source Data file.

response, as expected, was induced by CC-SpT-OTI/SnT-TRP2 OMVs (Fig. 6a, b). The analysis of IFNγ production and the levels of IFNγ$^+$ cytotoxic T lymphocytes showed that OVA$_{257-264}$ and TRP2$_{180-188}$ were both successfully presented to T lymphocytes when displayed on CC OMVs (groups 8 and 9; Fig. 6c–e); this effect was much stronger in the OMVs formulations than it was in the mixed formulations (groups 5 and 6, respectively). More importantly, the CC OMVs displaying two antigens induced more antigen-specific T lymphocytes than those displaying a single antigen (Fig. 6c–e and Supplementary Fig. 19). Together,

these data indicate that the display and presentation of multiple antigens to CD8$^+$ T cells can be realized using our platform. This strategy enables the coverage of complex and heterogeneous tumor antigens in vaccine design.

**Anti-tumor effects in a subcutaneous tumor model.** The subcutaneous MC38 tumor model was adopted to evaluate the anti-tumor immunity in the colon cancer model. Adpgk, a neoantigen in MC38 tumors, was labeled with SpT (SpT-Adpgk). SpT-Adpgk was

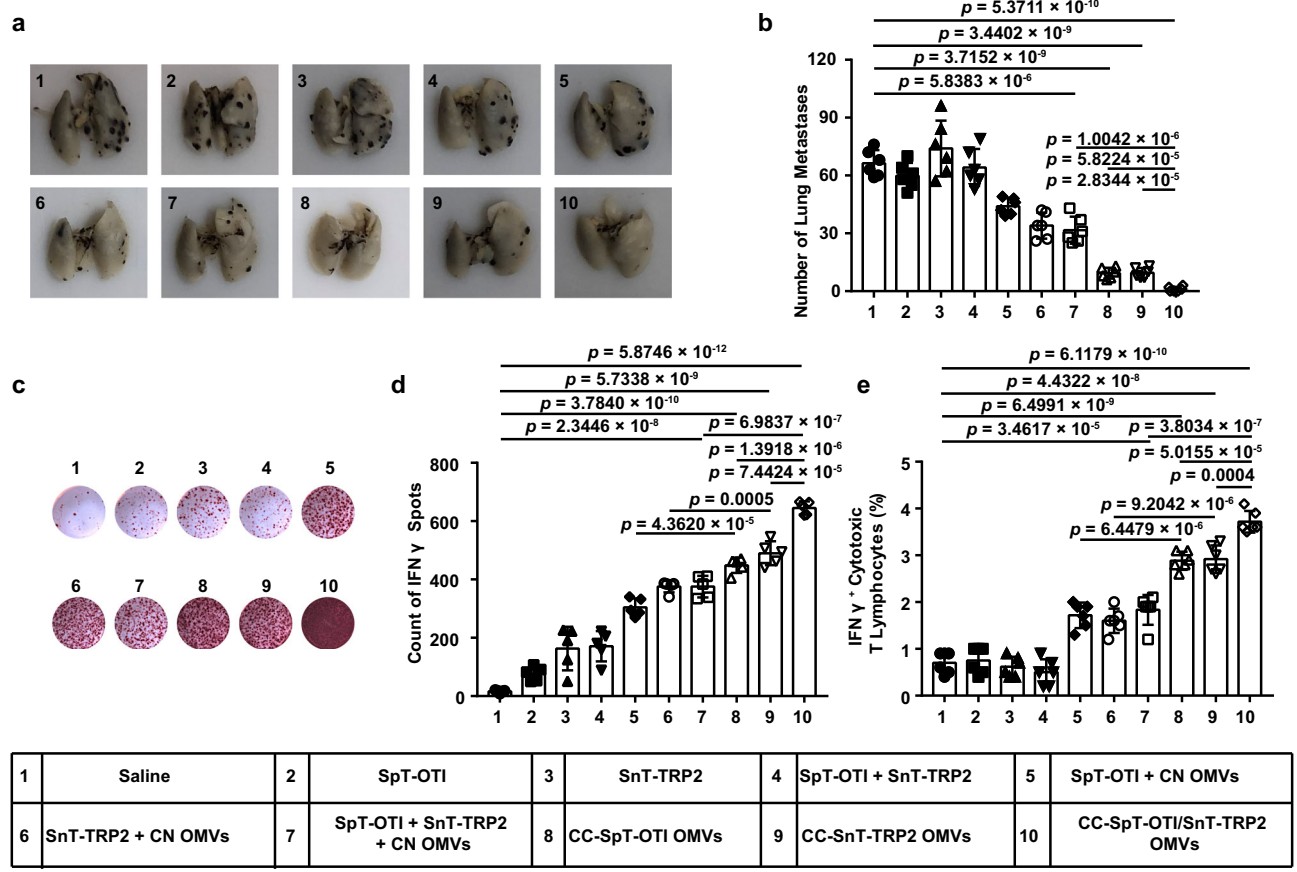

**Fig. 6 Dual-tumor antigen (OVA$_{257-264}$ and TRP2$_{180-188}$) display by catcher-decorated OMVs stimulates CD8$^+$ T-cell-mediated, synergistic immune therapeutic effects.** SpT-OTI and SnT-TRP2 were displayed either singly (CC-SpT-OTI OMVs or CC-SnT-TRP2 OMVs) or simultaneously (CC-SpT-OTI/ SnT-TRP2 OMVs) by CC OMVs. Mixed formulations (SpT-OTI + CN OMVs, SnT-TRP2 + CN OMVs, and SpT-OTI + SnT-TRP2 + CN OMVs) were used as controls. **a, b** Lungs were collected on day 17 at the end of the treatment period and photographed (**a**); the tumor nodules were counted (**b**). **c, d** IFNγ secretion by splenocytes isolated from the various treatment groups was determined by the ELISPOT assay after re-stimulation of the cells with OVA$_{257-264}$ and TRP2$_{180-188}$ (**c**). Quantitative data derived from the ELISPOT assays are shown in (**d**). **e** Flow cytometry analysis of IFNγ$^+$ cytotoxic T lymphocytes in splenocytes isolated from the various treatment groups and re-stimulated with OVA$_{257-264}$ and TRP2$_{180-188}$ antigens. Data are presented as the percentage of IFNγ$^+$ cells in the CD3$^+$CD8$^+$ T-cell subpopulation. The data (**b, e** ($n = 6$), **d** ($n = 5$)) are shown as mean ± SD. Statistical analysis was performed by a two-tailed unpaired $t$ test. Source data are provided as a Source Data file.

connected to CC OMVs to generate CC-SpT-Adpgk OMVs. The approved adjuvant Poly (I:C) and CN OMVs were used as a control to mix with SpT-Adpgk, respectively (Poly (I:C) + SpT-Adpgk and SpT-Adpgk + CN OMVs). The mice were vaccinated on days 3, 7, and 11 after tumor cell inoculation (Fig. 7a). Although the mixture formulations inhibited tumor growth slightly, CC-SpT-Adpgk OMVs exhibited the strongest inhibition effects on tumor growth (Fig. 7c and Supplementary Fig. 20). On day 50, 70% of mice in CC-SpT-Adpgk group still survived, which was much more than the survivors in Poly (I:C) + SpT-Adpgk group (30%). Meanwhile, all mice died before day 43 in saline and SpT-Adpgk + CN OMVs groups (Fig. 7c). More importantly, CC-SpT-Adpgk OMVs treatment led to complete regression of the tumors in 60% of the mice (Fig. 7d). In the treatment process, the tumors were harvested on day 29 and further digested into a single-cell suspension for cytometry analysis of immune cell infiltration. As expected, the infiltration of CD3$^+$ T cells, CD3$^+$CD8$^+$ T cells, CD3$^+$CD4$^+$ T cells, activated neutrophils (CD11b$^+$Ly6G$^+$ cells), and DCs (CD11c$^+$ cells) were all significantly elevated in MC38 tumor tissues after *s.c.* immunization with the CC-SpT-Adpgk OMVs (Fig. 7e and Supplementary Fig. 21). The immunosuppressive microenvironment mediated by regulatory T cells (Treg, CD3$^+$CD4$^+$Foxp$^+$ T cells) was alleviated effectively by CC-SpT-Adpgk OMVs treatment

(Fig. 7e). These immunomodulation effects in CC-SpT-Adpgk OMVs group were more dramatic than those in the mixture groups. There was no significant change in the infiltration of macrophages (F4/80$^+$ cells) in the tumor tissue between different groups. Interestingly, there was an infiltration of myeloid-derived suppressor cells (MDSCs, CD11b$^+$Gr$^+$ cells) after CC-SpT-Adpgk OMVs treatment, although the MDSCs infiltration did not disturb the antitumor effect (Fig. 7e). These results suggested that the OMVs-based platform is applicable for broad tumor types.

**The long-term immune memory in vivo elicited by antigen-loaded CC OMVs.** Successful induction of immune memory is critical for the long-term benefit of tumor vaccine. For immune memory studies, the mice were vaccinated on days 0, 3, and 8 (Fig. 8a). The vaccine formulations were shown in Fig. 8b. SpT-OVA (OVA$_{257-264}$) was connected to CC OMVs to generate CC-SpT-OVA OMVs. Control formulations included two mixture formulations (Poly (I:C) + SpT-OVA and SpT-OVA + CN OMVs). The splenocytes on days 60 were obtained and evaluated their cytotoxic effects on B16-OVA and MC38 cells. As shown in Fig. 8c and Supplementary Fig. 22a, the splenocytes exhibited a greater cytotoxic effect against B16-OVA cells in CC-SpT-OVA

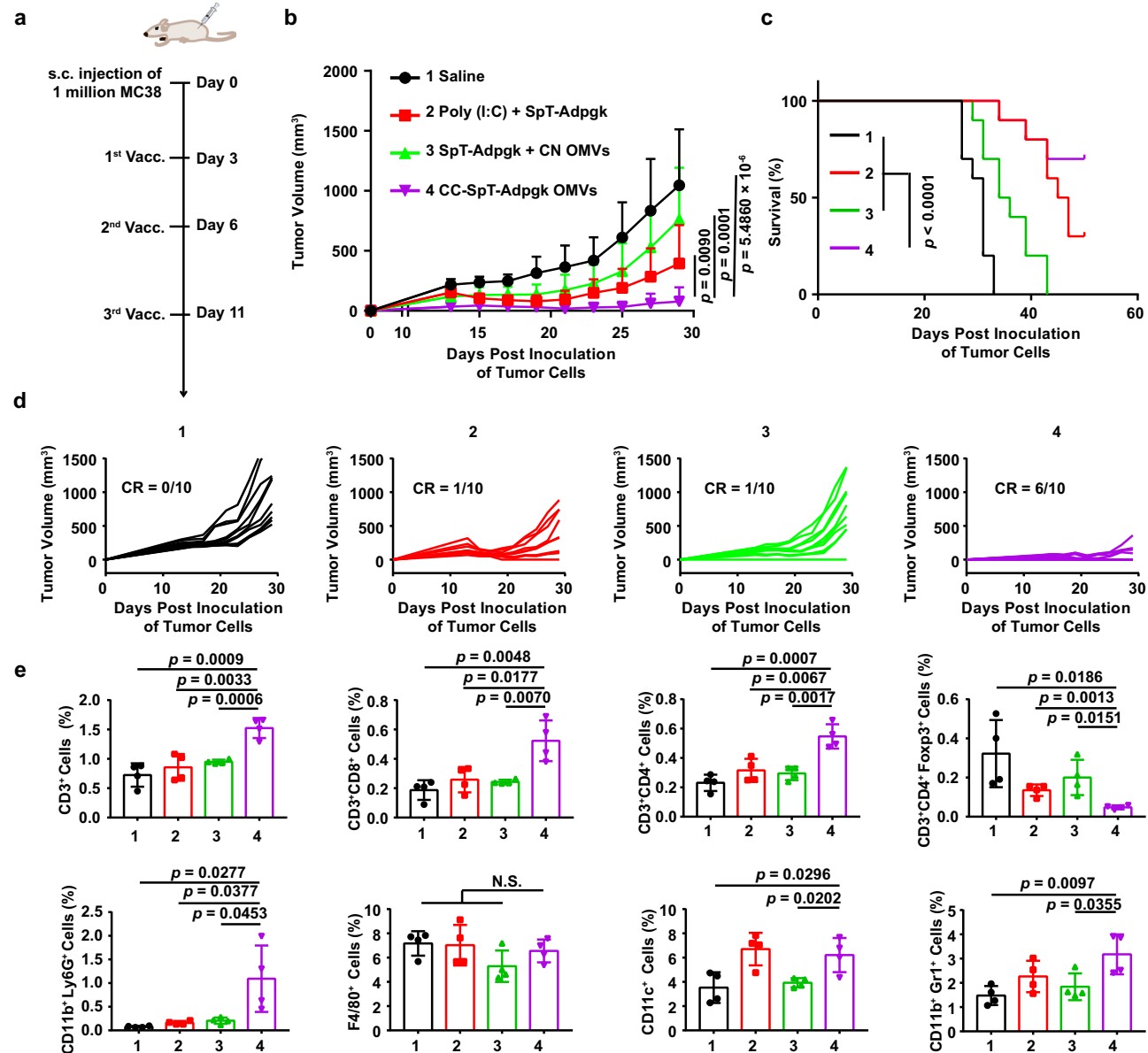

**Fig. 7 The anti-tumor immunity of the antigen-displayed OMVs in the subcutaneous MC38 tumor model.** The SpT-labeled Adpgk (a neoantigen in MC38 cells), SpT-Adpgk was displayed by CC OMVs (CC-SpT-Adpgk OMVs). The adjuvant Poly (I:C) and CN OMVs were used as a control to mix with SpT-Adpgk, respectively (Poly (I:C) + SpT-Adpgk and SpT-Adpgk + CN OMVs). **a** Schema showing the subcutaneous tumor model utilizing MC38 cells and the timing of vaccination (Vacc.) with different formulations. C57BL/6 mice were inoculated with MC38 cells ($1 \times 10^6$ cells/mouse, s.c.) and immunized with the indicated formulations on days 3, 7, and 11. **b–d** Tumor volumes were recorded, and survival was monitored. The data are shown as mean ± SD ($n = 10$). **e** In another set of MC38 tumor-bearing animals, tumors were harvested on day 29 for flow cytometry analysis ($n = 4$) of the following immune cells: $CD3^+$, $CD3^+CD8^+$, $CD3^+CD4^+$, $CD3^+CD4^+Foxp3^+$ T lymphocytes, activated neutrophils ($CD11b^+Ly6G^+$ cells), macrophages ($F4/80^+$ cells), dendritic cells ($CD11c^+$ cells), and MDSCs ($CD11b^+Gr1^+$ cells). The data are shown as mean ± SD. Statistical analysis was performed by two-tailed unpaired *t* test (**b**, **e**) and two-sided log-rank test (**c**). N.S. no significance. Source data are provided as a Source Data file.

OMVs group than that in other groups. This effect disappeared in the experiments using MC38 cells without OVA antigen, which provide robust evidence for the antigen specificity of the immune response (Fig. 8d and Supplementary Fig. 22b). Next, we quantified antigen-specific T cells in splenocytes and blood by flow cytometry. CC-SpT-OVA OMVs vaccination elicited more antigen-specific T cells (tetramer+ T cells) and IFNγ+ cytotoxic T lymphocytes than Poly (I:C) + SpT-OVA and SpT-OVA + CN OMVs, respectively (Fig. 8e–g and Supplementary Figs. 23a, b and 24). Furthermore, CC-SpT-OVA OMVs induced obvious central memory T-cell (~24%) and effector memory T cells (~7%) (Fig. 8h and Supplementary Fig. 25a, b) for over 60 days,

indicating that CC-SpT-OVA OMVs could be used as a prophylactic vaccine. On day 60, the immunized mice were challenged with i.v. injection of $2 \times 10^5$ B16-OVA cells. In contrast to the obvious lung metastasis in the mice immunized with mixture formulations, there was almost no lung metastasis in mice in CC-SpT-OVA OMVs group (Fig. 8i, j).

To further investigate the immunological memory, we test the tumor rechallenge in vaccine-cured mice. The mice were inoculated with B16-OVA cells and treated with CC-SpT-OVA OMVs vaccine. Then, the survived animals were rechallenged with s.c. injection of B16-F10 or B16-OVA on day 60 (Fig. 8k). CC-SpT-OVA OMVs protected 50% of mice from the B16-OVA cells

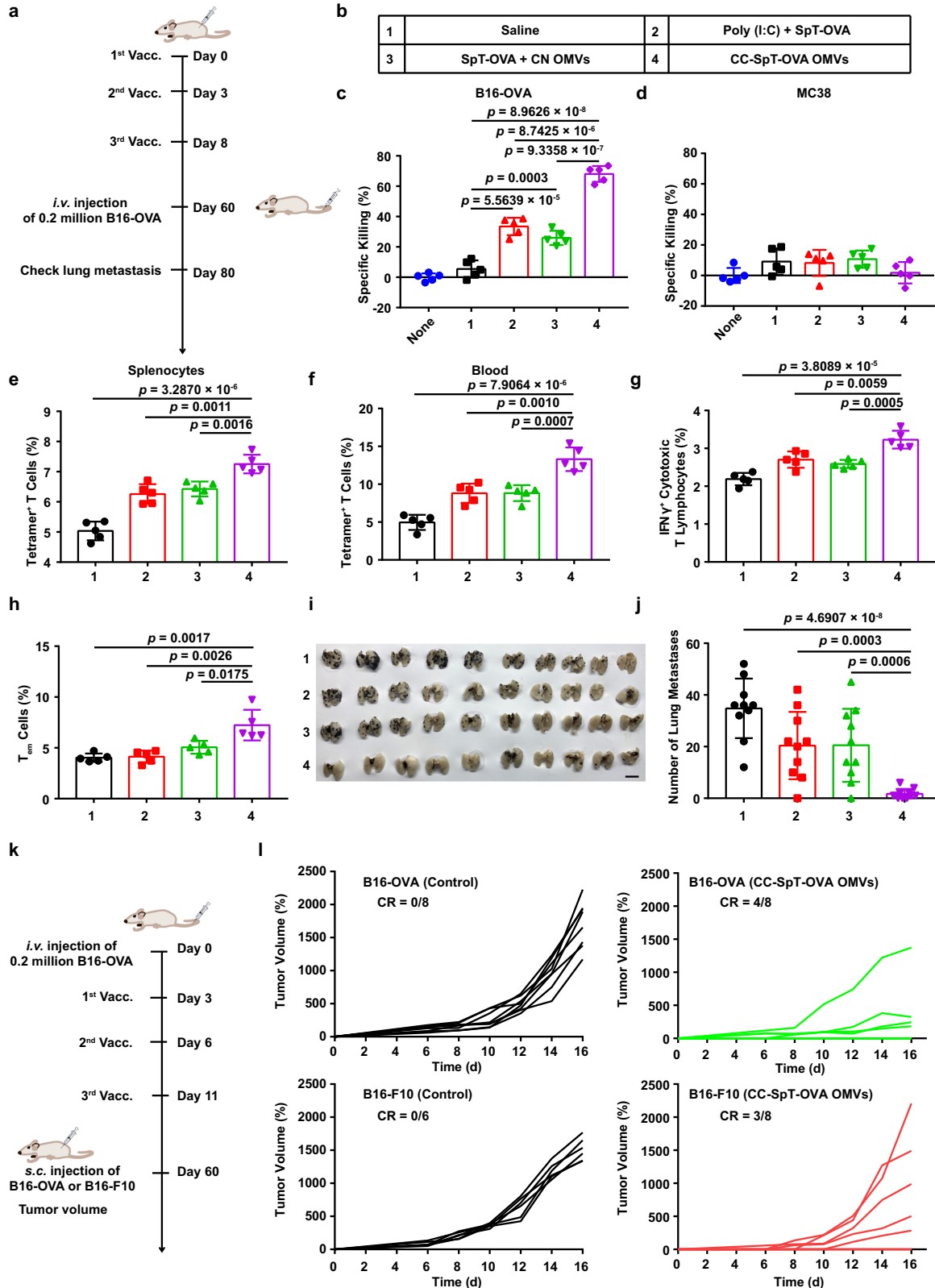

rechallenge (Fig. 8l). In addition, 37.5% of mice exhibited complete tumor resistance against B16-F10 cells rechallenge (Fig. 8k). These results indicated that CC-SpT-OVA OMVs can stimulate the antigen-specific immune memory, and the antigen released from the vaccine-killed tumor cells also induces an antigen-spreading immunity.

## Discussion

Once the host immune system is successfully presensitized by tumor neoantigens successfully, the tumor-specific T cells are activated and eliminate tumor cells. However, as described previously, the spontaneous processing and presentation of neoantigens by the immune system is very inefficient. The two most

**Fig. 8 The long-term immune memory in vivo elicited by antigen-loaded CC OMVs. a** Schema of immune memory analysis. C57BL/6 mice were immunized with the formulations shown in (**b**) on days 0, 3, and 8. Immune responses were evaluated, and the mice were challenged with B16-OVA cells ($2 \times 10^5$ cells/mouse, i.v.) on days 60. Lung metastasis was analyzed on day 80. **c, d** Specific killing ability of splenocytes collected on day 60 toward B16-OVA cells with OVA antigen (**c**) and MC38 cells without OVA antigen (**d**) analyzed by CCK-8 assay. **e, f** Quantitative analysis of tetramer$^+$ T cells in splenocytes (**e**) and blood (**f**) on day 60 through flow cytometry. **g** Flow cytometry analysis of IFNγ$^+$ cytotoxic T lymphocytes in splenocytes re-stimulated with OVA$_{257-264}$. **h** The proportion of T$_{em}$ cells (CD8$^+$CD44$^+$CD62L$^-$) in splenocytes on day 60 ($n = 5$). **I, j** Lungs were collected on day 80 and photographed ($n = 10$). **k** Schema of tumor rechallenge model. The mice were inoculated with B16-OVA cells and treated with CC-SpT-OVA OMVs vaccine. Then, the survived animals (complete tumor regression) were rechallenged with s.c. injection of B16-F10 and B16-OVA cells on day 60. **l** B16-OVA or B16-F10 tumor growth curve (B16-OVA (Control), $n = 8$; B16-OVA (CC-SpT-OVA OMVs), $n = 8$; B16-F10 (Control), $n = 6$; B16-F10 (CC-SpT-OVA OMVs), $n = 8$). The controls were healthy mice without tumor burden. The data are shown as mean ± SD. Statistical analysis was performed by a two-tailed unpaired $t$ test. Source data are provided as a Source Data file.

common ways to improve the immunogenicity of an antigen is to enhance the lymph node drainage of the antigen with vectors and to activate innate immunity with an immune adjuvant. Nanoparticles have proven to be effective vaccine vectors; the popular design of nanoparticle-based vaccines (nanovaccines) is to load the nanoparticle with adjuvant components[39,40]. OMVs are nature-endowed nanovaccine vectors with inherent adjuvant functionality. Our results show that OMVs can efficiently drain and deliver antigens into lymph nodes. At the same time, OMVs elicited excellent immune adjuvant functions, including stimulating DC maturation and promoting cytokine release. The underlying mechanisms of OMVs' adjuvant effect is that the LPS, lipoprotein, and flagellin in the OMVs can effectively activate TLR4 and the inflammasome, TLR2, and TLR5, respectively[41–44].

The next question is how to load OMVs with tumor neoantigens. Methods of exogenous antigen decoration on OMVs have been reported, including loading exogenous antigens onto the surface or into the lumen of OMVs by means of genetic engineering or electroporation, respectively[45]. In consideration of the rapid display by the tag/catcher pairs, we attempted a surface modification of OMVs via genetic engineering. The common surface scaffold proteins on OMVs include ClyA, hemoglobin protease (Hbp), and outer membrane protein (Omp) A/C/F[45]. There is a size limitation of surface display protein by Hbp[45]. The N- and C-terminals of OmpA/C/F are both on the medial side of the outer membrane[46–48], while the insertion of exogenous elements into the middle of the protein may affect the protein structure and correct folding, especially for the transmembrane protein. Therefore, ClyA was selected as the anchor site in our study, and the display of exogenous antigens on the surface of OMVs was realized by fusion expression at the C-terminal of ClyA. Although using ClyA to display microbial antigens or model antigens has previously been reported[22,49], our results validate that the tumor antigens displayed at the ClyA site can induce a strong antigen-specific anti-tumor immune response.

Neoantigens produced by somatic mutations in tumor cells must be presented to the immune system via forming complexes with MHCI molecules[4]. Although tumor cells possess MHCI molecules, there are no co-stimulation signals (CD80 and CD86), which are necessary for the final maturation of cytotoxic T cells[50]. Therefore, the delivery of tumor antigens to the professional antigen-presenting cells (mainly DCs) is important for effective anti-tumor immunity. However, the antigens on OMVs are foreign antigens for DCs. In general, foreign antigens are presented to stimulate CD4$^+$ T cells via the MHCII-mediated pathway, not CD8$^+$ T cells which are primarily responsible for the anti-tumor immune response. In order to interact with MHCI molecular to activate antigen-specific CD8$^+$ T cells, the foreign antigens need to enter the cytoplasm via a process called cross-presentation, in which endosomal escape after antigen endocytosis is a key step[34]. The endosomal escape of OMVs and their cargoes (including LPS and antigens) has been reported previously[41]. In consideration of

the rapid degradation of free antigen peptide in the endosome and lysosome, the co-delivery of antigen and OMVs to a single DC is crucial for successful antigen presentation. This is a major reason for the higher MHCI-antigen complexes in DCs and the stronger anti-tumor immunity in the ligation system than those in the mixed system in our results. Another potential mechanism for this phenomenon is that the more favorable and efficient uptake of OMVs into cells, when compared to antigens, causes a premature differentiation of DCs which leads to inefficient antigen processing. By linking antigens using a Plug-and-Display system, our nanovaccine ensures not only a rapid antigen display but also efficient antigen processing and presentation.

Rapid antigen display on vaccine vector has received increasing attention. Such mechanisms have been reported in vaccines against pathogenic microorganisms for rapid vaccine production during epidemic outbreaks[30,51]. As mentioned above, the heterogeneity of neoantigens in and between tumor cells also requires a rapid antigen display technology in the tumor vaccine. The current tumor nanovaccine we describe here loads the personalized antigens into nanovectors. However, the all-in-one antigen encapsulation into the nanoparticles during the production process hinders the ability to freely select and match antigens in the anti-tumor application. By employing the Plug-and-Display technology, the vector and antigen in our nanovaccine were separately synthesized, requiring only a simple combination procedure before immunization. This modular design allows us to establish a neoantigen library in advance and rapidly select appropriate antigen combinations, which may reduce the production time and realize the bedside preparation of tumor vaccines for individual patients in the future.

By expressing the protein Plug-and-Display system in conjunction with ClyA on the surface of OMVs, we have developed a flexible system for the display of tumor antigens. Due to the efficient OMVs accumulation into the lymph nodes and the stable integration of the antigen with the OMVs, antigens can be efficiently delivered to lymph nodes and presented to DCs. We used this OMV-based platform to deliver a series of tumor antigens in murine tumor models and demonstrated excellent antigen-specific T-lymphocyte-mediated anti-tumor immune responses. This bioengineered OMVs system was able to simultaneously display multiple tumor antigens, an approach that may be valuable in the development of personalized tumor vaccines that target complex and heterogeneous tumor antigens.

## Methods

**Materials**. DAPI was obtained from Life Technologies (Shanghai, China); anti-HA-Tag (catalog no. ab236632, dilution: 1:1000), anti-firefly luciferase monoclonal antibody (catalog no. ab16466, dilution: 1:10,000), and HRP-conjugated goat anti-mouse IgG (catalog no. ab205719, dilution: 1:5000) were purchased from Abcam (UK). Horseradish peroxidase (HRP) conjugated goat anti-rabbit IgG (catalog no. A16096, dilution: 1:5000) was bought from Invitrogen (USA). FITC-anti-mouse CD3 (catalog no. 100204, dilution: 1:50), APC-anti-mouse CD8α (catalog no. 100712, dilution: 1:80), PE/Cy7-anti-mouse IFNγ (catalog no. 505826, dilution:

1:20), PE-anti-mouse IFNγ (catalog no. 505808, dilution: 1:80), FITC-anti-mouse CD11c (catalog no. 117306, dilution: 1:200), PE/Cy7-anti-mouse CD80 (catalog no. 104734, dilution: 1:40), APC-anti-mouse CD86 (catalog no. 105012, dilution: 1:80), PE/Cy7-anti-mouse H-2Kb bound to SIINFEKL (PE/Cy7-anti-mouse MHCI-OVA) (catalog no. 114616, dilution: 1:40), PE-anti-mouse H-2Kb bound to SIINFEKL (PE-anti-mouse MHCI-OVA) (catalog no. 114608, dilution: 1:80), PE-anti-mouse Foxp3 (catalog no. 126404, dilution: 1:20), PE-anti-mouse CD44 (catalog no. 103008, dilution: 1:80), PE/Cy7-anti-mouse CD62L (catalog no. 104418, dilution: 1:80), PE/Cy7-anti-mouse F4/80 (catalog no. 123114, dilution: 1:80), APC-anti-mouse Gr1 (catalog no. 108412, dilution: 1:80), APC-anti-mouse CD11b (catalog no. 101212, dilution: 1:80), PE/Cy7-anti-mouse CD8 (catalog no. 100722, dilution: 1:80), FITC-anti-mouse CD4 (catalog no. 100406, dilution: 1:200), PE-anti-mouse CD11b (catalog no. 101208, dilution: 1:80) and FITC-anti-mouse Ly6G (catalog no. 127606, dilution: 1:200) were purchased from BioLegend (USA). FITC-anti-mouse CD8 (catalog no. 2002714, dilution: 1:100) and PE-anti-mouse CD4 (catalog no. 2013481, dilution: 1:160) were purchased from Invitrogen (USA). T-Select H-2Kb OVA Tetramer-SIINFEKL-PE (catalog no. TS-5001-1C, dilution: 1:5) was purchased from MBL Beijing Biotech Co., Ltd. The tumor necrosis factor α (TNF-α) mouse uncoated ELISA kit (catalog no. 88-7324-86), IL-6 mouse uncoated ELISA kit (catalog no. 88-7064-86), and IL-1β mouse uncoated ELISA kit (catalog no. 88-7013-86) were purchased from eBioscience (USA). The mouse IFNγ precoated ELISPOT kit (catalog no. 2210005) was purchased from Dakewe Biotech Co., Ltd (Shenzhen, China). OVA$_{257-264}$ (SIINFEKL), OVA$_{323-339}$ (ISQAVHAAHAEINEAGR), TRP2$_{180-188}$ (SVYDFFVWL), SpT-OTI (VPTIVMVDAYKRYKGGSIINFEKL), SnT-OTII (GKLGDIEFIKVNKGYGGISQAVHAAHAEINEAGR), SnT-TRP2 (GKLGDIEFIKVNKGYGGSVYDFFVWL), SpT-OVA-Cy5.5 (VPTIVMVDAYKRYKGGSIINFEKL-Cy5.5), Adpgk (CGIPVHLELASMTNMELMSSIVHQQVFPT), and SpT-Adpgk (VPTIVMVDAYKRYKGGCGIPVHLELASMTNMELMSSIVHQQVFPT) were synthesized by Top Peptide (Shanghai, China) via Fmoc solid-phase peptide synthesis. Confocal microscopy Petri dishes (Hangzhou Xinyou Biotechnology Co., Ltd., China) were purchased from HuaLiDe Technology Co., Ltd. Hochest 33342 (catalog no. C0030) was purchased from Solarbio Life Science (Beijing, China). According to the quality reports provided by the manufacturer, the purity of all peptides was over 95%.

**Animals and cells**. Female C57BL/6 mice (6–8-week-old) were purchased from Vital River Laboratory Animal Technology Co. Ltd (Beijing, China). Mice were housed in a room with a temperature of 20–22 °C and a humidity of 30–70%. Feed and water were available ad libitum. Artificial light was provided in a 12-h light/12-h dark cycle. This study complied with relevant ethical regulations for animal testing and research, all animal protocols were approved by the Institutional Animal Care and Use Committee of the National Center for Nanoscience and Technology. The murine melanoma (B16-F10), murine colon cancer (MC38), murine pancreatic cancer (Pan 02), and DC2.4 cell lines were obtained from the ATCC (Manassas, USA). B16-OVA cells were generously provided by P. Wang Hao at the National Center for Nanoscience and Technology. B16-F10, MC38, Pan 02, and B16-OVA cells were cultured in DMEM containing 10% fetal bovine serum, 100 U mL$^{-1}$ penicillin G sodium, 100 µg mL$^{-1}$ streptomycin (Pen/Strep), and 20 µM β-mercaptoethanol (β-ME). DC2.4 cells were cultured in RPMI-1640 medium supplemented with 10% fetal bovine serum, 100 U mL$^{-1}$ penicillin G sodium, and 100 µg mL$^{-1}$ streptomycin (Pen/Strep). All cell lines were tested mycoplasma-free. The cells were incubated at 37 °C in a humidified environment with 5% CO$_2$. All cell culture medium and FBS were purchased from Wisent (Canada).

**Plasmid construction, bacterial strain, and growth**. The genes encoding ClyA-OVA-3HA, Luciferase, ClyA-Luciferase, and ClyA-None were cloned into pET28a (Genewiz, Suzhou, China). ClyA-Catcher was cloned into pETDuet-1. The E. coli OMVs production strain Rosetta (DE3) (Tiangen Biotech Beijing Co., Ltd, China, wild-type LPS) which had been transformed with the expression plasmids pET28a-ClyA-OVA-3HA, pET28a-ClyA-Luciferase or pETDuet-1-ClyA-Catcher was grown at 37 °C in LB medium, with shaking at 180 rpm, until the OD$_{600}$ was 0.6. isopropyl β-D-1-thiogalactopyranoside (IPTG, 0.1 mM) was added to further induce protein expression at 16 °C, and incubation was continued for 14 h with shaking at 160 rpm. Kanamycin (50 ug/µL) or ampicillin (50 µg/mL) was added when appropriate.

**OMVs purification and characterization**. Briefly, E. coli were cultured as described above, then removed by centrifugation at 5000×g for 10 min at 4 °C. The resulting supernatant (200 mL) was filtered through a 0.45-µm EPS filter (Millipore), then concentrated to 50 mL using a 50-K ultrafiltration tube. The concentrated solution was further filtered with a 0.22-µm EPS membrane (Millipore). OMVs were collected from the filtrate by ultracentrifugation at 150,000×g for 3 h at 4 °C. The collected OMVs were washed with PBS using centrifugation at 150,000×g for 2 h at 4 °C, then finally resuspended in 400 µL PBS and stored at −20 °C until use. The total protein concentration of OMVs preparations was evaluated using the bicinchoninic acid assay, the results of which were defined as the OMVs WT concentration. The size and morphology of the OMVs were characterized using dynamic light scattering (DLS) (Zetasizer Nano ZS90, Malvern, UK) and transmission electron microscopy (TEM) (Tecnai G2 F20 U-TWIN, FEI, USA). The LPS content in OMV was detected by ELISA (CEB526Ge, Cloud-Clone Corp., Wuhan, China) and LAL assay (L00350C, GenScript, Nanjing, China).

**Western blot analysis**. Total bacterial protein was extracted using the Bacterial Protein Extraction Kit (Beijing Puyihua Science and Technology Co., Ltd., China, catalog no. C600596). Briefly, bacteria were collected and washed by centrifugation at 5000×g for 10 min at 4 °C. The bacteria pellets were resuspended in 400 µL 1 × cell lysis buffer containing 4 µL phenylmethanesulfonyl fluoride (PMSF; Solarbio, China) and 80 µl lysozyme per 1 ml bacterial culture. The suspension was incubated at 37 °C for 30 min. In order to fully lyse the bacteria, the mixture was further incubated on a rocking platform for 10 min. In total, 20 µl DNaseI/RNase was added to the mixture, which was then incubated with rocking for another 10 min at 37 °C. The insoluble debris was removed by centrifugation at 3000×g for 30 min at 4 °C. The supernatant was collected for further use. For the protein extraction of OMVs, the OMVs were resuspended in RIPA buffer (Solarbio, China) containing 1 mM PMSF and lysed for 15 min. After centrifugation at 13,000×g for 15 min, the supernatant was collected. Protein samples were electrophoresed on sodium dodecyl sulfate-polyacrylamide gels (10% polyacrylamide), and then transferred onto a polyvinylidene fluoride membrane. The membranes were blocked in 10% nonfat milk and subsequently incubated with an anti-firefly luciferase monoclonal antibody or anti-HA-Tag antibody for 2 h at room temperature. After washing three times using Tris-buffered saline containing 0.5% Tween 20 (Solarbio, China), the membrane was incubated with HRP-conjugated goat anti-rabbit IgG or HRP-conjugated goat anti-mouse IgG for 1 h at room temperature. Immunoreactive proteins were visualized using Super Signal West Pico Chemiluminescent Substrate (Thermo Scientific, Rockford, USA).

**In vitro BMDC maturation, cellular uptake experiments, and cross-presentation assays**. Bone marrow cells were flushed from the femurs and tibias of C57BL/6 mice and cultured in RPMI-1640 supplemented with 10% FBS, 100 U mL$^{-1}$ penicillin G sodium, 100 µg mL$^{-1}$ streptomycin, 1% HEPES, 0.05 mM β-ME, 20 ng/mL IL-4 and 20 ng/mL GM-CSF after the red blood cells had been lysed. The cultures were initiated by placing ~1–1.5 × 10$^6$ bone marrow cells per well into six-well plates. Half the medium was replaced every 2 days. On day 6, non-adherent cells were collected for further investigation. To assess their maturation state, murine BMDCs were cultured with PBS, peptide antigen (50 µg/mL), or a mixture of peptide antigen and the different OMVs formulations (50 µg/mL) in a 1.5-mL tube for 24 h. At the completion of the incubation, the cells were collected for further staining with FITC-anti-mouse CD11c, PE/Cy7-anti-mouse CD80, or APC-anti-mouse CD86 to evaluate BMDC maturation. For the cross-presentation assay, BMDCs, DC2.4, or Pan 02 were incubated with OVA (or SpT-OVA) alone or a mixture of OVA (or SpT-OVA) and different OMVs formulations at 37 °C for 3–24 h. The OVA$_{257-264}$ presentation by MHCI on the cell surface was then detected using a PE/Cy7-anti-mouse H-2Kb bound to SIINFEKL antibody, and the fluorescence was detected by flow cytometry (BD Accuri C6, BD Biosciences, USA) or laser scanning confocal microscopy (LSCM, Zeiss LSM710, Germany). To assess cellular uptake, SpT-OVA-Cy5.5 or a mixture of SpT-OVA-Cy5.5 and different OMVs formulations (50 µg mL$^{-1}$) was incubated with murine BMDCs on glass-bottom culture dishes at 37 °C for 6 or 12 h, then examined using LSCM.

**Cytokine assay**. Bone marrow-derived dendritic cells (BMDCs) from C57BL/6 mice were cultured at 1 × 10$^6$ cells/well in 200 µL complete media mentioned before transferring them to 96-well, round-bottomed plates with OVA$_{257-264}$ alone or a mixture of OVA$_{257-264}$ and different OMVs formulations at 37 °C for 3–24 h. The supernatant was harvested at 3, 6, 12, and 24 h. TNF-α, IL-1β, and IL-6 were analyzed with ELISA kits according to the manufacturer's protocols.

**Cytotoxicity assay**. BMDCs were seeded into 1.5-mL tubes at a density of 1 × 10$^5$ cells/tube in the presence of different concentrations of OMVs. After treatment for 24 h, the cells were collected and stained with Annexin V-APC and 7-AAD at room temperature for 30 min. Then cells were washed three times with PBS before analysis by flow cytometry.

**Luciferase detection and ex vivo imaging**. Fluorescein potassium was added to the bacteria (after they had been treated with IPTG) and their secreted OMVs (addition of ATP before detection). The samples were prepared for bioluminescence and measured using a Maestro in vivo spectrum imaging system (IVIS; Cambridge Research & Instrumentation, Woburn, MA, USA).

Prior to ex vivo imaging, saline, SpT-OVA-Cy5.5, or a mixture of SpT-OVA-Cy5.5 and different OMVs were subcutaneously injected into the tail base of mice for the evaluation of their biodistribution. At various time points thereafter, the inguinal lymph nodes and major organs, including the heart, liver, spleen, lung, and kidney, were collected for ex vivo fluorescence examination using the Maestro system. Subsequently, the inguinal lymph nodes were flash-frozen and cryosectioned (10-µm sections). The sections were stained with a DAPI nuclear stain and examined by LSCM.

**Immunization and tumor therapy experiments**. C57BL/6 mice (6–8 weeks old; n ≥ 4 for each group) were injected intravenously with B16-OVA or B16-F10 melanoma cells ($2 \times 10^5$). Animals used for evaluating the anti-tumor effects of single-tumor antigen-displayed OMVs were immunized by subcutaneous injection into the tail base using different OMVs formulations (50 μg per antigen peptide; 50 μg OMVs) or controls on days 3, 6, and 11 after the administration of the tumor cells. For the anti-tumor effects of dual-tumor antigens displayed OMVs, the mice were immunized twice with the different formulations on days 3 and 7 after the administration of the tumor cells. On day 17, the mice were euthanized and the lungs were collected, briefly rinsed with PBS, and fixed with Fekete's buffer (70 mL 75% alcohol, 10 mL formalin, and 5 mL glacial acetic acid). After 48 h fixation, the lungs were photographed and the pulmonary tumor nodules were counted.

Splenocytes were isolated from the spleen of immunized mice for intracellular IFNγ staining and IFNγ ELISPOT analysis. Briefly, the peptide antigen was co-cultured with the splenocytes overnight. Splenocytes treated with ionomycin (BioGems, USA, Catalog no. 5608212) were used as positive controls. Monensin (BioGems, USA, Catalog no. 2237803) was added to the cells 5 h before staining. The cells were collected for intracellular staining following the protocol provided by the manufacturer (Intracellular Flow Cytometry Staining Protocol provided by BioLegend). The surface markers CD3 and CD8 were stained prior to fixation, and the cells were fixed by a commercially available fixation buffer (BioLegend, USA, Catalog no. 420801) and permeabilized prior to IFNγ staining. The cells were then washed with intracellular staining perm wash buffer (BioLegend, USA, Catalog no. 421002) and resuspended in cell staining buffer for flow cytometry analysis (BD Accuri C6, BD Biosciences, USA). For IFNγ ELISPOT assays, all operations were carried out in accordance with the protocol provided by the manufacturer. Briefly, splenocytes were seeded at $1 \times 10^5$ cells per well in a 96-well plate coated with a mouse anti-IFNγ antibody and incubated for 20 h with peptide antigen or ionomycin (positive control). The secreted and captured IFNγ was subsequently detected using a biotinylated antibody specific for IFNγ and alkaline-phosphatase conjugated to streptavidin. After the addition of the substrate solution, a brown precipitate formed and appeared as spots at the sites of cytokine production. Automated spot quantification was performed by Dakewe Biotech Co., Ltd.

A subcutaneous colon cancer model was constructed for evaluating the therapeutic effect of nanovaccine in solid tumors. In total, $1 \times 10^6$ MC38 cells were inoculated into the right-back of mice on days 0. The mice were immunized with different formulations subcutaneously (Saline, 50 μg Poly (I:C) + 50 μg SpT-Adpgk, 50 μg SpT-Adpgk + 50 μg CN OMVs, 50 μg CC-SpT-Adpgk OMVs) on days 3, 6, and 11 (n = 10/group). Tumor volume was recorded every other day and calculated by the following equation: tumor volume = length × width$^2$ × 0.5. Mice were sacrificed and tumors were collected on days 29. Tumors were then digested into a single-cell suspension for evaluating the infiltration of immune cells (n = 4). The survival for each mouse was recorded for plotting the survival curve in another set of subcutaneous MC38 tumor model. The survival experiment was terminated when tumor volume reached 1500 mm$^3$ (n = 10).

**Immune memory**. Female C57BL/6 mice (6–8 weeks, n = 5) were vaccinated on days 0, 3, and 8. Peripheral blood and splenocytes was collected on day 60 to analyze antigen-specific CD8$^+$ T cells (tetramer$^+$ T cells and IFNγ$^+$ cytotoxic T-lymphocyte cells) and memory T cells (T$_{naive}$ (CD3$^+$CD8$^+$CD44$^-$CD62L$^+$), T$_{cm}$ (CD3$^+$CD8$^+$CD44$^+$CD62L$^+$), T$_{em}$ (CD3$^+$CD8$^+$CD44$^+$CD62L$^-$)) by flow cytometry analysis.

Splenocytes were further used for the specific killing assays. Splenocytes were cultured with antigen (OVA$_{257-264}$) overnight and then cultured with B16-OVA cells and MC38 cells at the ratio of 10:1 for 24 h. Non-adherent cells were removed, and adherent cells were washed with PBS. CCK assay was used to calculate the percent of specific killing.

For prophylactic tumor challenge studies, vaccinated mice (n = 10) were challenged on day 60 by intravenous injection of $2 \times 10^5$ B16-OVA cells per mouse. On day 80, the mice were euthanized and the lungs were collected, briefly rinsed with PBS, and fixed with Fekete's buffer. After 48 h fixation, the lungs were photographed, and the pulmonary tumor nodules were counted.

**Immune memory in tumor rechallenge model**. Female C57BL/6 mice were injected intravenously with $2 \times 10^5$ B16-OVA cells on day 0. Mice were immunized by subcutaneous injection with CC-SpT-Adpgk OMVs on days 3, 6, and 11. Survived mice were further subcutaneously inoculated with $1 \times 10^6$ B16-OVA cells or $1 \times 10^6$ B16-F10 cells per mouse on the right flank on days 60 (n = 6–8). Then, tumor volume was recorded every other day and calculated by the same equation mentioned before.

**Immunohistochemical analysis of T-lymphocyte infiltration into lungs**. The lungs were harvested when treatments were terminated, fixed in paraformaldehyde, and paraffin-embedded. Lung sections (7 μm) were deparaffinized and rehydrated. The sections were treated with 3% H$_2$O$_2$ at room temperature for 10 min to eliminate the activity of endogenous peroxidase. After antigen retrieval in 10 mM citrate buffer (pH 6.0) at 95 °C for 15 min, the sections were blocked with 5% goat serum/PBS for 1 h and incubated with anti-CD8 (Abcam, UK, catalog no. ab93278,

dilution: 1:100) at 4 °C, overnight, followed by incubation with goat anti-rabbit IgG biotinylated antibody (Biorbyt, UK, catalog no. orb153693, dilution: 1:100) at room temperature for 1 h and then HRP-conjugated streptavidin at 37 °C for 30 min. DAB was utilized for color development. Sections were counterstained with hematoxylin. Images were obtained using an Olympus BX 51 microscope (Olympus).

**Statistical and reproducibility**. Data are presented as the mean ± SD. At least three independent experiments were performed for each in vitro study (Fig. 1a, c-d; 3b-g, i; Supplementary Fig. 1, 2a–c, 9, 10a–c, 11a–c, 12, 14a-b, 15b, 17c-d, 22a-b). Statistical analysis was carried out using SPSS version 19.0. Data were analyzed by a two-tailed unpaired t test for comparison of two groups. Kaplan–Meier curves were analyzed using two-sided log-rank tests. Origin Pro 8.5.1, GraphPad Prism 5, BD Accuri C6 Software, FlowJo V10, and ImageJ v1.8.0 were used to analyze the acquired data.

## Data availability

All the other data supporting the findings of this study are available within the article and its Supplementary Information file and from the corresponding author upon reasonable request. Source data are provided with this paper.

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

## Acknowledgements

This work was supported by the grants from National Key R&D Program of China (2018YFA0208900 and 2018YFE0205300), Beijing Natural Science Foundation of China (Z200020), Beijing Nova Program (Z201100006820031), the National Natural Science Foundation of China (31800838, 31820103004, 31730032, 31800799 and 51861145302), the Key Research Project of Frontier Science of the Chinese Academy of Sciences (QYZDJ-SSW-SLH022), the Innovation Research Group of National Natural Science Foundation (11621505), and the Hundred-Talent Program of Chinese Academy of Sciences.

## Author contributions

K.C. and R.Z. contributed equally to this work. G.N., X.Z., K.C., and R.Z. designed the research. K.C., Y.L. (Yao Li), Y.Q., Y.W., Y.Z., H.Q., Y.Q., L.C., J.L., Y.L. (Yujing Li), J.X., and J.S. performed the research. All authors analyzed and interpreted the data. K.C., X.Z., L.R., G.N., and G.A. wrote the paper.

## Competing interests

The authors G.N., X.Z., and K.C. filed a patent based on this technology (Patent application number: CN202011407688.7. Title: "Bacterial outer membrane vesicle, a general nanovaccine containing bacterial outer membrane vesicle, preparation method and application thereof". The patent is currently under review). The remaining authors declare no competing interests.
