## [Peer Review File · Nature Communications]

REVIEWER COMMENTS

Reviewer #3 (Remarks to the Author):

The manuscript describes experiments showing how OMVs from E.coli can be used to display tumor antigens, and in such a way that protection against melanoma metastasis can be demonstrated in a mouse model. The study is well written and thorough and the experiments are well documented. Overall it is an important addition to the OMV-vaccine field.

Specific comments:

1. The ClyA fusion proteins are said to be surface-exposed, but I don't see the evidence for that. Only localization to the OMVs is demonstrated, not surface localization as with the plug-and-display system.
2. The OMV platform is not described in sufficient detail. What are the genetic characteristics of the E.coli strain? Does it have wildtype or mutant LPS? Mutations to increase OMV formation? Given the toxicity of LPS, this is an important question. Will the OMVs not be too reactogenic/toxic for human applications?

Reviewer #4 (Remarks to the Author):

This MS of Cheng...Nie presents a new approach to vaccination using OMV nanoparticles of GN bacteria and a ClyA-fusion system for capture of one or more antigens for presentation. The authors suggest both induction of innate and adaptive immune responses in a murine test system, and support this with data. They do not adequately describe the ClyA system and this ought to be addressed. While their focus is upon developing an approach to tumor antigens, and specifically neoantigens, the question arises at this time due to pandemic regarding suitability of this approach to SARS CoV 2. It would be of interest to see studies upon human solid tumors to address the clinical question that confronts oncology, but this clearly lies outside the domain of the authors' work.

A smaller issue is whether functional effects of F/T were assessed beyond the morphological effects reported L211.

There are multiple typographical and syntactic issues (article 'the' omitted L32) and innate composition (that has no immediate interpretation for this reviewer L64). There are multiple areas where the symbol □ 'box' is inserted in place of '?' L 119 and 'ff' L147, L546)

Reviewer #5 (Remarks to the Author):

The work of Dr. Cheng et al. describes a versatile Outer Membrane Vesicle (OMV)-based vaccine platform to elicit a specific anti-tumor immune response. They show that bioengineered OMVs can efficiently display and present different tumor antigens, inducing a strong antigen specific immune response that subsequently result in abrogating lung melanoma metastasis through T cell-mediated immunity. In addition, OMVs decorated with different protein catchers can simultaneously display multiple, distinct tumor antigens to elicit a synergistic anti-tumor immune response. The ability of our bioengineered OMV-based platform rapidly and simultaneously display antigens may facilitate the application of these agents in the development of personalized tumor vaccines.

The work is innovative and interesting from the bioengineering point of view but in my opinion suffers important points from the cancer immunology point of view.

Main Points

- 1) The tumor models used in this paper are simplistic and do not robustly test the efficacy of the system. TRP and OVA antigens are highly immunogenic peptides that their efficacy has been showed in combination with an appropriate adjuvant. I would like to see an experiment where this system has been tested with different neo-antigens and tumor models other than the B16 family. In addition the lung metastasis model is interesting but survival curves and tumor growth would also be needed to complete the study.
- 2) A more in-depth immunological analysis would be needed to evaluate what kind of immunity the OMV are eliciting, only T cells? Only CD8 T cells? What kind of phenotype these T cells show? Are they all effector? What about memory in comparison with more classical vaccine approach (Poly-IC + antigen for example).
- 3) I believe this system works really well because of the ability of the OMVs to travel to lymph nodes efficiently hence explaining why OMV conjugates with the peptides resulted better than the mixture of them. However, to fully elucidate the mode of action it would be necessary to see a group of mice treated with Poly IC and the peptides (or other adjuvants).
- 4) The vaccination effect of this system was not very thoroughly tested. Two types of experiments could have been done. First type of experiment would be vaccinating the mice with the OMV system and then testing the engraftment of the tumor. Second type of experiment would be re-challenging the mice with the same tumor. Abscopal effect could have been also a useful model to shed light on the mechanism.
- 5) The immunological analyses of the tumor microenvironment are lacking. It would be interesting to see if there is an increase of CD8+ T cells and/or CD4+ T cells, DC, neutrophils or macrophages infiltrating the tumor
- 6) Why the authors are using only ELISPOT (please indicate on the y axes the amount of splenocytes tested, this info is only in material and methods) to check the presence of T cells specific response? TRP2 or SIINFEKL tetramer do exist.

Minor Comments

- 1) Supplementary figure 3, cell viability assay with different methods would have been beneficial. What about 7AAD and Annexin-V to further show this
- 2) Line 147, there are two f's missing in effect
- 3) In the first mice experiment with the OVA-model, an increase of CD4+ T cells is shown but this could be an increase of Tregs to counterbalance the increased immunity. Authors should examine that this increase is not a T-reg increase but rather a Th1 response (since it's a bacterial component)
- 4) In Supplementary figure 10, increase the FBS percentage to at least 50% to show how robust this method is and imitating in vivo setting as close as possible
- 5) Supplementary figure 13, in the CD80 and CD86+ upregulation experiment it would be beneficial to add also CN- OMVs alone and CO-OVA just to check whether the linkage of this SpT/SpC pair and SnT/SnC pair affect DC maturation
- 6) In Figure 3G why is the SpT-OVA-Cy5.5 condition showing more presentation compare to the CC-SpT-OVA-Cy5.5 OMVs? please discuss

7) In Figure 3h, why were the formulations injected intradermal?? Wouldn't it be better to do SC as done previously in the efficacy tests? please discuss

8) In Fig4g, why is there more inf-gamma secreted with CN OMVs compared to SnT-TRP2? It seems like the OMVs have TRP2 peptides or other peptides with high homology?

Point-by-point responses to reviewers

Note: Following are our responses (in blue color) to reviewers' comments (in bold black color) and sentences described in the revised manuscript (highlighted in yellow).

Responses to Reviewer 3

The manuscript describes experiments showing how OMVs from E.coli can be used to display tumor antigens, and in such a way that protection against melanoma metastasis can be demonstrated in a mouse model. The study is well written and thorough and the experiments are well documented. Overall it is an important addition to the OMV-vaccine field.

Specific comments:

1. The ClyA fusion proteins are said to be surface-exposed, but I don't see the evidence for that. Only localization to the OMVs is demonstrated, not surface localization as with the plug-and-display system.

Response 1: Thank you very much for the comment. The site of the fusion protein should avoid affecting the protein structure as much as possible. As a transmembrane protein, the suitable modification sites of ClyA are N-terminal and C-terminal. In this study, we fused the catchers to the C-terminal of ClyA, which is common in the previous studies¹⁻³. We confirmed the location of catchers through their reaction with the gold nanoparticles-labelled tags (**Figure 3e**).

We also tried to fuse the catchers into the N-terminal of ClyA. According to the structure of ClyA (**Figure R1 and R2**)^{4, 5}, both N-terminal and C-terminal of ClyA are extracellular. As shown in **Figure R3**, either N-terminal or C-terminal fusion with SpyCatcher on ClyA can bind SpyTag, indicated by the bands at the molecular weight of about 45 kDa.

Figure R1. Three-dimensional structure of ClyA⁴.

Figure R2. The structure of ClyA protein on membrane⁵.

Figure R3. The conjugation of SpT-HA to ClyA-SpyCatcher was verified by western blot analysis using an anti-HA antibody. The ClyA-Catcher-N and ClyA-Catcher-C indicated N-terminal and C-terminal fusion with SpyCatcher on ClyA, respectively.

2. The OMV platform is not described in sufficient detail. What are the genetic characteristics of the *E.coli* strain? Does it have wildtype or mutant LPS? Mutations to increase OMV formation? Given the toxicity of LPS, this is an important question. Will the OMVs not be too reactogenic/toxic for human applications?

Response 2: The *E.coli* strain used in the study is Rosetta (DE3), which is the popular stain in genetic engineering and has wildtype LPS expression. We detected that the level of LPS in the OMVs (1 mg/mL) was 120 ng/ml. Compared with the half lethal dose (LD₅₀) of LPS (300 μg/mouse), the dose of OMVs (50 μg OMV per mouse, 6 ng LPS per mouse) we used in this study is significantly lower. We also evaluated the effects of OMVs on cell activity *in vitro*. The data show that OMVs were biocompatible and non-toxic at all concentrations tested when incubated with BMDCs cells *in vitro* (new **Supplementary Figure 3**). The biosafety of the OMVs has also been confirmed in clinic. The OMVs-based Group B meningococcal vaccine MeNZB has effectively limited the incidence and mortality of meningitis in New Zealand.

The new **Supplementary Figure 3** was added into manuscript as follows:

New Supplementary Figure 3. The cytotoxicity of OMVs in murine bone marrow-derived dendritic cells (BMDCs) was measured by flow cytometry after 24 h incubation with CO OMVs at the indicated protein concentrations or PBS. Cells were stained with annexin V-APC/7-AAD. The Annexin V⁻/7-AAD⁻ cells are viable cell.

Responses to Reviewer 4

This MS of Cheng...Nie presents a new approach to vaccination using OMV nanoparticles of GN bacteria and a ClyA-fusion system for capture of one or more antigens for presentation. The authors suggest both induction of innate and adaptive immune responses in a murine test system, and support this with data. They do not adequately describe the ClyA system and this ought to be addressed. While their focus is upon developing an approach to tumor antigens, and specifically neoantigens, the question arises at this time due to pandemic regarding suitability of this approach to SARS CoV2. It would be of interest to see studies upon human solid tumors to address the clinical question that confronts oncology, but this clearly lies outside the domain of the authors' work.

Thank you very much for the positive comment. Cytolysin A (ClyA) is a pore-forming toxin synthesized by *Escherichia coli* and other enteric bacteria. ClyA has been used to display microbial antigens or model antigens in the previous studies^{6,7}. As a transmembrane protein, the suitable modification sites of ClyA are N-terminal and C-terminal. In this study, we fused the catchers to the C-terminal of ClyA, which is commonly used in the previous studies for the surface modification of OMVs. In this study, the OMVs-based vaccine platform was mainly used to induce T-cell-mediated immunity. Several studies showed that T-cell-mediated immunity plays an important role in COVID-19 prevention and treatment⁸⁻¹⁰.

1. A smaller issue is whether functional effects of F/T were assessed beyond the morphological effects reported L211.

Response 1: Thank you for the helpful comment. We have assessed the function of frozen-thawing CC-SpT-OVA OMVs in the revised manuscript. After 12 h of co-culture with different formulations (PBS, fresh CC-SpT-OVA OMVs or repetitive freezing-thawing CC-SpT-OVA OMVs), BMDCs were collected for flow cytometry analysis for maturation (CD11c⁺CD80⁺CD86⁺) and antigen presentation (CD11c⁺MHC I-OVA⁺). The results were shown in the new **Supplementary Figure 10d-e**, which was added into manuscript as follow:

After conjugating with SpT-OVA, repetitive freezing-thawing CC-SpT-OVA OMVs could also effectively stimulate the maturation of BMDCs (**Supplementary Figure 10d**) and promote the presentation of antigens (**Supplementary Figure 10e**), with no significant difference compare to the non-freezing-thawing (fresh) preparation.

New Supplementary Figure 10. (a)-(c) TEM images and DLS analysis of fresh CC OMVs (a), CC OMVs after 5 freeze-thaw cycles (-80 °C) (b) and CC OMVs after incubation in 10% FBS for 24 h (c). Scale bar, 100 nm. (d) and (e) The analysis of immune stimulation function of different formulation. The expression of the maturation markers CD80⁺CD86⁺ was examined as a percentage of CD11c⁺ cells by flow cytometry (d). The expression of the MHCI-OVA complex on the surface of BMDCs was measured by flow cytometry (CD11c⁺MHC I-OVA⁺) (e).

2. There are multiple typographical and syntactic issues (article 'the' omitted L32) and innate composition (that has no immediate interpretation for this reviewer L64). There are multiple areas where the symbol □ 'box' is inserted in place of '?' 'C' L 119 and 'ff' L147, L546)

Response 2: Thanks for your help. We have carefully read our manuscript and improved our description accordingly.

Responses to Reviewer 5

The work of Dr. Cheng et al. describes a versatile Outer Membrane Vesicle (OMV)-based vaccine platform to elicit a specific anti-tumor immune response. They show that bioengineered OMVs can efficiently display and present different tumor antigens, inducing a string antigen specific immune response that subsequently result in abrogating lung melanoma metastasis through T cell-mediated immunity. In addition, OMVs decorated with different protein catchers can simultaneously display multiple, distinct tumor antigens to elicit a synergistic anti-tumor immune response. The ability of our bioengineered OMV-based platform rapidly and simultaneously display antigens may facilitate the application of these agents in the development of personalized tumor vaccines.

The work is innovative and interesting from the bioengineering point of view but in my opinion suffers important points from the cancer immunology point of view.

Main Points

1) The tumor models used in this paper are simplistic and do not robustly test the efficacy of the system. TRP and OVA antigens are highly immunogenic peptides that their efficacy has been showed in combination with an appropriate adjuvant. I would like to see an experiment where this system has been tested with different neo-antigens and tumor models other than the B16 family. In addition the lung metastasis model is interesting but survival curves and tumor growth would also be needed to complete the study.

Response 1: Thank you very much for the constructive comment. According to your suggestion, we further used the OMVs-based vaccine platform to display and deliver a real tumor neoantigen, Adpgk, and evaluated the anti-tumor immunity in the subcutaneous MC38 tumor model. The results were shown in new **Figure 7**, and were added into the manuscript as follows:

Anti-tumor effects in a subcutaneous tumor model

The subcutaneous MC38 tumor model was adopted to evaluate the anti-tumor immunity in the colon cancer model. Adpgk, a neoantigen in MC38 tumors, was labeled with SpT (SpT-Adpgk). SpT-Adpgk was connected to CC OMVs to generate CC-SpT-Adpgk OMVs. The approved adjuvant Poly (I:C) and CN OMVs were used as control to mix with SpT-Adpgk, respectively [Poly (I:C) + SpT-Adpgk and SpT-Adpgk + CN OMVs]. The mice were vaccinated on days 3, 7 and 11 after tumor cell inoculation (**Figure 7a**). Although the mixture formulations inhibited tumor growth slightly, CC-SpT-Adpgk OMVs exhibited the strongest inhibition effects on tumor growth (**Figure 7c and Supplementary Figure 20**). At days 50, 70% mice in CC-SpT-Adpgk group were still survived, which was much more than the survivors in Poly (I:C) + SpT-Adpgk group (30%). Meanwhile, all mice died before days 43 in saline and SpT-Adpgk + CN OMVs groups (**Figure 7c**). More importantly, CC-SpT-Adpgk OMVs treatment led to complete regression of the tumors in 60% of the mice (**Figure 7d**). During the treatment process, the tumors were harvested on days 29 and further digested into single cell suspension for cytometry analysis of immune cell infiltration. As expected, the infiltration of CD3⁺ T cells, CD3⁺CD8⁺ T cells, CD3⁺CD4⁺ T cells, activated neutrophils (CD11b⁺Ly6G⁺ cells) and DCs (CD11c⁺ cells) were all significantly elevated in MC38 tumor tissues after *s.c.* immunization with the CC-SpT-Adpgk OMVs (**Figure 7e and Supplementary Figure 21**). The immunosuppressive microenvironment mediated by regulatory T cells (Treg, CD3⁺CD4⁺Foxp⁺ T cells) was alleviated effectively by CC-SpT-Adpgk OMVs treatment (**Figure 7e**). These immunomodulation effects in CC-SpT-Adpgk OMVs group were more dramatic than that in the mixture groups. There was no significant change in the infiltration of macrophages (F4/80⁺ cells) in the tumor tissue between different groups. Interestingly, there was an infiltration of myeloid-derived suppressor cells (MDSCs, CD11b⁺Gr⁺ cells) after CC-SpT-Adpgk OMVs treatment, although the MDSCs infiltration did not disturb the anti-tumor effect (**Figure 7e**). These results suggest that the OMVs-based platform is applicable for broad tumor types.

Figure 7. The anti-tumor immunity of the antigen-displayed OMVs in the subcutaneous MC38 tumor model. The SpT-labelled Adpgk (a neoantigen in MC38 cells), SpT-Adpgk was displayed by CC OMVs (CC-SpT-Adpgk OMVs). The adjuvant Poly (I:C) and CN OMVs were used as control to mix with SpT-Adpgk, respectively [Poly (I:C) + SpT-Adpgk and SpT-Adpgk + CN OMVs]. (a) Scheme showing the subcutaneous tumor model utilizing MC38 cells and the timing of vaccination (Vacc.) with different formulations. C57BL/6 mice were inoculated with MC38 cells (1×10^6 cells/mouse, *s.c.*) and immunized with the indicated formulations on days 3, 7 and 11. (b)-(d) Tumor volumes were recorded, and survival was monitored. The data were shown as mean \pm s.d. ($n = 10$). (e) In another set of MC38 tumor-bearing animals, tumors were harvested on days 29 for flow cytometry analysis ($n = 10$) of the following immune cells: $CD3^+$, $CD3^+CD8^+$, $CD3^+CD4^+$, $CD3^+CD4^+Foxp3^+$ T lymphocytes, activated neutrophils ($CD11b^+Ly6G^+$ cells), macrophages ($F4/80^+$ cells), dendritic cells ($CD11c^+$ cells) and MDSCs ($CD11b^+Gr1^+$ cells). N.S., no significance; *, $P < 0.05$; **, $P < 0.01$; ***, $P < 0.001$.

2) A more in-depth immunological analysis would be needed to evaluate what kind of immunity the OMV are eliciting, only T cells? Only CD8 T cells? What kind of phenotype these T cells show? Are they all effector? What about memory in comparison with more classical vaccine approach (Poly-IC + antigen for example).

Response 2: We greatly appreciate the thoughtful suggestion. In this study, we mainly focus on antigen-specific immune responses. The immunity induced by OMV-antigen depends on the type of antigens. We found that CC-MHC I- restricted epitopes OMVs (CO OMVs, CC-SnT-TRP2 OMVs and CC-SpT-OTI OMVs) could elicit a strong increase in the number of IFN γ ⁺ cytotoxic T lymphocytes (CD3⁺CD8⁺IFN γ ⁺, **Figure 2l, 4f, 5e and 6e**). When the introduced antigens were MHC II-restricted epitopes (CC-SnT-OTII OMVs), elevated proportions of CD3⁺CD4⁺IFN γ ⁺ cells were found (**Figure 5f**).

We systemically studied the immune memory, and the results were added into the revised manuscript as follows:

The long-term immune memory *in vivo* elicited by antigen-loaded CC OMVs

Successful induction of immune memory is critical for long-term benefit of tumor vaccine. For immune memory studies, the mice were vaccinated on days 0, 3 and 8 (**Figure 8a**). The vaccine formulations were shown in **Figure 8b**. SpT-OVA (OVA₂₅₇₋₂₆₄) was connected to CC OMVs to generate CC-SpT-OVA OMVs. Control formulations included two mixture formulations (Poly (I:C) + SpT-OVA and SpT-OVA + CN OMVs). The splenocytes on days 60 were obtained and evaluated their cytotoxic effects on B16-OVA and MC38 cells. As shown in **Figure 8c and Supplementary Figure 22a**, the splenocytes exhibited a greater cytotoxic effect against B16-OVA cells in CC-SpT-OVA OMVs group than that in other groups. This effect disappeared in the experiments using MC38 cells without OVA antigen, which provide robust evidence for the antigen specificity of immune response (**Figure 8d and Supplementary Figure 22b**). Next, we quantified antigen-specific T cells in splenocytes and blood by flow cytometry. CC-SpT-OVA OMVs vaccination elicited more antigen-specific T cells (tetramer⁺ T cells) and IFN γ ⁺ cytotoxic T lymphocytes than Poly (I:C) + SpT-OVA and SpT-OVA + CN OMVs, respectively (**Figure 8e-8g and Supplementary Figure 23a, 23b and 24**). Furthermore, CC-SpT-OVA OMVs induced obvious central memory T cell (~24%) and effector memory T cells (~7%) (**Figure 8h and Supplementary Figure 25a, 25b**) for over 60 days, indicating that CC-SpT-OVA OMVs could be used as prophylactic vaccine. On days 60, the immunized mice were challenged with *i.v.* injection of 2×10^5 B16-OVA cells. In contrast to the obvious lung metastasis in the mice immunized with mixture formulations, there was almost no lung metastasis in mice in CC-SpT-OVA OMVs group (**Figure 8i and 8j**).

To further investigate the immunological memory, we test the tumor rechallenge in vaccine-cured mice. The mice were inoculated with B16-OVA cells and treated with CC-SpT-OVA OMVs vaccine. Then, the survived animals were rechallenged with *s.c.* injection of B16-F10 or B16-OVA on days 60 (**Figure 8k**). CC-SpT-OVA OMVs protected 50% mice from the B16-OVA cells rechallenge (**Figure 8l**). In addition, 37.5% mice exhibited complete tumor resistance against B16-F10 cells rechallenge (**Figure 8k**). These results indicated that CC-SpT-OVA OMVs can stimulate the antigen-specific immune memory, and the antigen released from the vaccine-killed tumor cells also induce an antigen-spreading immunity.

Figure 8. The long-term immune memory *in vivo* elicited by antigen-loaded CC OMVs. (a) Schema of immune memory analysis. C57BL/6 mice were immunized with the formulations shown in (b) on days 0, 3 and 8. Immune responses were evaluated, and the mice were challenged with B16-OVA cells (2×10^5 cells/mouse, *i.v.*) on days 60. Lung metastasis was analyzed on days 80. (c) and (d) Specific killing ability of splenocytes collected on days 60 toward B16-OVA cells with OVA antigen (c) and MC38 cells without OVA antigen (d) analyzed by CCK-8 assay. (e)

and (f) Quantitative analysis of tetramer⁺ T cells in splenocytes **(e)** and blood **(f)** on days 60 through flow cytometry. **(g)** Flow cytometry analysis of IFN γ ⁺ cytotoxic T lymphocytes in splenocytes re-stimulated with OVA₂₅₇₋₂₆₄. **(h)** The proportion of T_{em} cells (CD8⁺CD44⁺CD62L⁻) in splenocytes on days 60. **(i) and (j)** Lungs were collected on days 80 and photographed. **(k)** Schema of tumor re-challenge model. The mice were inoculated with B16-OVA cells and treated with CC-SpT-OVA OMVs vaccine. Then, the survived animals (complete tumor regression) were rechallenged with *s.c.* injection of B16-F10 and B16-OVA cells on days 60. **(l)** B16-OVA or B16-F10 tumor growth curve. The controls were healthy mice without tumor burden. The data were shown as mean \pm s.d. (n = 6-8). *, $P < 0.05$; **, $P < 0.01$; ***, $P < 0.001$.

3) I believe this system works really well because of the ability of the OMVs to travel to lymph nodes efficiently hence explaining why OMV conjugates with the peptides resulted better than the mixture of them. However, to fully elucidate the mode of action it would be necessary to see a group of mice treated with Poly IC and the peptides (or other adjuvants).

Response 3. Thank you for your helpful suggestion. We have added two animal experiments to compare the anti-tumor efficacy of our vaccine platform and mixed formulation (SpT-antigen + Poly (I:C)), which consists of approved adjuvants and antigen. Compared to the Poly (I:C), the OMVs-based platform induced stronger anti-tumor immunity in the subcutaneous MC38 tumor model (new **Figure 7**) and stimulated more immune memory cells to protect the mice from following tumor cells challenge (new **Figure 8**).

4) The vaccination effect of this system was not very thoroughly tested. Two types of experiments could have been done. First type of experiment would be vaccinating the mice with the OMV system and then testing the engraftment of the tumor. Second type of experiment would be re-challenging the mice with the same tumor. Abscopal effect could have been also a useful model to shed light on the mechanism.

Response 4. Thank you for the insightful comment. We have completed these two types of experiments, which were shown as new **Figure 8**. Vaccination using our OMVs-based platform stimulated significant immune memory and protected the mice from the following tumor cells challenge. In addition, 50% vaccine-cured mice exhibited complete tumor resistance against the tumor cells re-challenge.

5) The immunological analyses of the tumor microenvironment are lacking. It would be interesting to see if there is an increase of CD8⁺ T cells and/or CD4⁺ T cells, DC, neutrophils or macrophages infiltrating the tumor.

Response 5. We have completed the immunological analyses of the tumor

microenvironment in the subcutaneous MC38 tumor model. The results were shown as new **Figure 7** as follows:

As expected, the infiltration of CD3⁺ T cells, CD3⁺CD8⁺ T cells, CD3⁺CD4⁺ T cells, activated neutrophils (CD11b⁺Ly6G⁺ cells) and DCs (CD11c⁺ cells) were all significantly elevated in MC38 tumor tissues after *s.c.* immunization with the CC-SpT-Adpgk OMVs (**Figure 7e and Supplementary Figure 21**). The immunosuppressive microenvironment mediated by regulatory T cells (Treg, CD3⁺CD4⁺Foxp⁺ T cells) was alleviated effectively by CC-SpT-Adpgk OMVs treatment (**Figure 7e**). These immunomodulation effects in CC-SpT-Adpgk OMVs group were more dramatic than that in the mixture groups. There was no significant change in the infiltration of macrophages (F4/80⁺ cells) in the tumor tissue between different groups. Interestingly, there was an infiltration of myeloid-derived suppressor cells (MDSCs, CD11b⁺Gr⁺ cells) after CC-SpT-Adpgk OMVs treatment, although the MDSCs infiltration did not disturb the anti-tumor effect (**Figure 7e**).

6) Why the authors are using only ELISPOT (please indicate on the y axes the amount of splenocytes tested, this info is only in material and methods) to check the presence of T cells specific response? TRP2 or SIINFEKL tetramer do exist.

Response 6. Thank you for your helpful suggestion. We have included the amount of splenocytes tested on Y axes in **Figure 2n, 4h, 5d and 6d**, according to the suggestions. We also supplemented SIINFEKL tetramer detection experiment in the revised manuscript (new **Figure 8e and 8f**).

Minor Comments

1) **Supplementary figure 3, cell viability assay with different methods would have been beneficial. What about 7AAD and Annexin-V to further show this.**

Response 1: We have optimized the experiment in the revised manuscript as follows:

We first confirmed that OMVs were not toxic to murine BMDCs in the concentration range used in the current study, using annexin V-APC/7-AAD apoptosis detection assay to stain dead cells (**Supplementary Figure 3**).

New Supplementary Figure 3. The cytotoxicity of OMVs in murine bone marrow-derived dendritic cells (BMDCs) was measured by flow cytometry after 24 h incubation with CO OMVs at the indicated protein concentrations or PBS. Cells were stained with annexin V-APC/7-AAD. The Annexin V/7-AAD⁻ cells are viable cell.

2) Line 147, there are two f's missing in effect.

Response 2: We have revised the sentences and carefully checked the whole manuscript.

3) In the first mice experiment with the OVA-model, an increase of CD4⁺ T cells is shown but this could be an increase of Tregs to counterbalance the increased immunity. Authors should examine that this increase is not a T-reg increase but rather a Th1 response (since it's a bacterial component).

Response 3: We greatly appreciate the constructive suggestions made by this reviewer. As shown in new **Figure 7e**, we found that the infiltration of CD3⁺CD4⁺ T cells were elevated and regulatory T cells (Treg, CD3⁺CD4⁺Foxp⁺ T cells) were alleviated effectively by the CC-SpT-Adpgk OMVs treatment, indicating CC-antigens OMVs alleviate the tumor immunosuppressive microenvironment.

4) In Supplementary figure 10, increase the FBS percentage to at least 50% to show how robust this method is and imitating in vivo setting as close as possible.

Response 4: According to the suggestion, CC OMVs were incubated with 50% fetal bovine serum (FBS) and the morphology was characterized by TEM. As the results shown in **Figure R4**, the background of TEM image is dirty due to high content of protein in PBS containing 50% FBS. Importantly, we still found that the morphology of CC OMVs was unaffected after 24 h incubation, suggesting that CC OMVs are likely to remain stable enough for vaccination to be effective.

Figure R4. TEM image of CC OMVs after incubation in 50% FBS for 24 h. Scale bar, 50 nm.

5) Supplementary figure 13, in the CD80 and CD86⁺ upregulation experiment it would be beneficial to add also CN-OMVs alone and CO-OVA just to check whether the linkage of this SpT/SpC pair and SnT/SnC pair affect DC maturation.

Response 5: As shown in **Figure 2a and 2b**, we measured the proportion of CD80⁺ and CD86⁺ in CD11c⁺ BMDCs cultured with CN OMVs, OVA₂₅₇₋₂₆₄ + CN OMVs or CO OMVs, and found that all three formulations induced a significant increase in the proportion of CD80⁺ and CD86⁺.

6) In **Figure 3G** why is the SpT-OVA-Cy5.5 condition showing more presentation compare to the CC-SpT-OVA-Cy5.5 OMVs? please discuss.

Response 6: Figure 3g showed the cell uptake results. The antigen presentation was shown in **Supplementary Figure 13c**. There was no significant difference in the amount of MHCII-OVA complex between SpT-OVA and CC-SpT-OVA OMVs. We hypothesize that OMV may promote the escape of the endosomes due to membrane fusion, thus enhancing cross-presentation.

7) In **Figure 3h**, why were the formulations injected intradermal?? Wouldn't it be better to do SC as done previously in the efficacy tests? please discuss.

Response 7. Thank you for pointing out this mistake. Animals used in this study were all immunized by subcutaneous injection into the tail base. We have revised the description and checked the whole manuscript.

8) In **Fig4g**, why is there more inf-gamma secreted with CN OMVs compared to

SnT-TRP2? It seems like the OMVs have TRP2 peptides or other peptides with high homology?

Response 8. It has been validated that OMVs suppress tumor growth by interferon- γ -mediated anti-tumor response¹¹. The killed tumor cells will release tumor antigens, including TRP2 for recognition by the immune system, resulting in an increase in IFN γ ⁺ cytotoxic T lymphocytes. In spite of the presence of the specific antigen in SnT-TRP2 group, the lack of adjuvant effect of OMV to stimulate innate immunity may lead to an inability to stimulate effective antigen specific immune response.

References

1. Gujrati, V. et al. Bioengineered Bacterial Outer Membrane Vesicles as Cell-Specific Drug-Delivery Vehicles for Cancer Therapy. *ACS Nano* **8**, 1525-1537 (2020).
2. Kim, J. Y. et al. Engineered Bacterial Outer Membrane Vesicles with Enhanced Functionality. *J. Mol. Biol.* **380**, 51-66 (2008).
3. Chen, D. J. et al. Delivery of Foreign Antigens by Engineered Outer Membrane Vesicle Vaccines. *Proc. Natl. Acad. Sci. U S A.* **107**, 3099-3104 (2010).
4. Gregor Anderluh, Jeremy Lakey, Gregor Anderluh. Proteins Membrane Binding and Pore Formation. *Springer New York*, 2010.
5. Mueller, M. et al. The structure of a cytolytic α -helical toxin pore reveals its assembly mechanism. *Nature* **459**, 726-730 (2009).
6. Rappazzo, C.G. et al. Recombinant M2e outer membrane vesicle vaccines protect against lethal influenza A challenge in BALB/c mice. *Vaccine* **34**, 1252-1258 (2016).
7. Schettters, S.T.T. et al. Outer membrane vesicles engineered to express membrane-bound antigen program dendritic cells for cross-presentation to CD8⁺ T cells. *Acta Biomater.* **91**, 248-257 (2019).
8. Moderbacher, C.R. et al. Antigen-Specific Adaptive Immunity to SARS-CoV-2 in Acute COVID-19 and Associations with Age and Disease Severity. *Cell* **183**, 1-17 (2020).
9. Grifoni, A. et al. Targets of T Cell Responses to SARS-CoV-2 Coronavirus in Humans with COVID-19 Disease and Unexposed Individuals. *Cell* **181**, 1489-1501 (2020).
10. Weiskopf, D. et al. Phenotype and kinetics of SARS-CoV-2-specific T cells in COVID-19 patients with acute respiratory distress syndrome. *Sci Immunol.* **5**, eabd2071 (2020).
11. Kim, O.Y. et al. Bacterial outer membrane vesicles suppress tumor by interferon- γ -mediated antitumor response. *Nat. Commun.* **8**, 626 (2017).

REVIEWER COMMENTS

Reviewer #3 (Remarks to the Author):

The authors have partially answered my questions:

1. Regarding the surface exposure of the constructs: Fig. 3e shows surface exposure (by EM immunostaining) of the ClyA-catcher constructs. However, my question was about the ClyA luciferase fusions which are described in the first paragraph.
2. Regarding the OMV characterization: more data have been added, including a cytotoxicity assay. However, the reference to the safety of the meningococcal OMV vaccine is not relevant, as this vaccine has been made in a different way: extraction with deoxycholate was used to remove most of the LPS. The OMVs of the present study contain wildtype LPS and no detergent extraction step, reactogenicity therefore may still be a problem.

Reviewer #5 (Remarks to the Author):

The authors have replied with scrupulosity and have addressed all the main points that concerned me.

Point-by-point responses to the reviewer

Note: Following are our responses (in blue color) to reviewers' comments (in bold black color) and sentences described in the revised manuscript are highlighted in yellow.

Reviewer 3

The authors have partially answered my questions:

1. Regarding the surface exposure of the constructs: Fig. 3e shows surface exposure (by TEM immunostaining) of the ClyA-catcher constructs. However, my question was about the ClyA luciferase fusions which are described in the first paragraph.

Response 1: According to the structure of ClyA, the suitable modification sites of ClyA are N-terminal and C-terminal, and both N-terminal and C-terminal of ClyA are extracellular^{1,2}. In this study, we chose the C-terminal of ClyA as the modification site. In addition, it is difficult for small molecules to enter OMVs without external stimulation or stressors³⁻⁶. Therefore, only when luciferase is displayed onto the surface of OMVs, it can react with its substrate to generate bioluminescence accordingly. We first showed the expression of ClyA-Luc in OMVs (**Figure 1a**). After adding the luciferase substrate, fluorescein potassium, emitted bioluminescence was detected immediately in the ClyA-Luc group only (**Figure 1b**). Those observation indicates that luciferase was displayed on the surface of OMVs.

2. Regarding the OMV characterization: more data have been added, including a cytotoxicity assay. However, the reference to the safety of the meningococcal OMV vaccine is not relevant, as this vaccine has been made in a different way: extraction with deoxycholate was used to remove most of the LPS. The OMVs of the present study contain wildtype LPS and no detergent extraction step, reactogenicity therefore may still be a problem.

Response 2: We are grateful for the reviewer to point of the importance of LPS level in the vaccine formulation. To address the issue, we evaluated the level of IFN γ and IL-12P70 in the serum of mice immunized with the formulations shown in **Figure 5**, OMVs did not cause the storm of inflammatory cytokines, indicating that the immune response stimulated by OMVs was within the safe range for mice (**Figure R1**). We did not see any obvious skin damage and granuloma at the injection site of the mice.

In this manuscript, we detected that the level of LPS in the wild-type OMVs (1 mg/ml) was 120 ng/ml. Compared with the half lethal dose (LD50) of LPS (300 $\mu\text{g}/\text{mouse}$), the dose of OMVs (50 μg OMV per mouse, 6 ng LPS per mouse) we used in this study is significantly lower. We monitored the mice body weight in the subcutaneous MC38 tumor model, and the OMVs we used did not cause significant weight loss in the mice immunized with CC-SpT-Adpgk OMVs and SpT-Adpgk + CN OMVs (**Figure R2**), further indicating that the wild OMVs were safe vaccine vectors.

Figure R1. The level of IFN γ (a) and IL-12P70 (b) in in the serum of mice immunized with the indicated formulations. The timing of vaccination (Vacc.) with the OMVs preparations was shown in **Supplementary Figure 17a**, and the serum was collected on days 17.

Figure R2. Body weight of all animals was recorded during each treatment, with all animals appearing healthy throughout the study based on eating and behavior.

References

1. Gregor Anderluh, Jeremy Lakey, Gregor Anderluh. Proteins Membrane Binding and Pore Formation. *Springer New York*, 2010.
2. Mueller, M. et al. The structure of a cytolytic α -helical toxin pore reveals its assembly mechanism. *Nature* **459**, 726-730 (2009).
3. Gerritzen, M.J.H., Martens, D.E., Wijffels, R.H., van der Pol, L. & Stork, M. Bioengineering bacterial outer membrane vesicles as vaccine platform. *Biotechnol. Adv.* **35**, 565-574 (2017).
4. Fuhrmann G, et al. Active loading into extracellular vesicles significantly improves the cellular uptake and photodynamic effect of porphyrins. *J. Control. Release*, **205**, 35-44 (2015).
5. Lamichhane T N, Raiker R S, Jay S M. Exogenous DNA Loading into Extracellular Vesicles via Electroporation is Size-Dependent and Enables Limited Gene Delivery. *Mol. Pharm.* **12**, 3650-3657 (2015).
6. Gujrati V, et al. Bioengineered Bacterial Outer Membrane Vesicles as Cell-Specific Drug-Delivery Vehicles for Cancer Therapy. *ACS Nano*, **8**, 1525-1537 (2014).

REVIEWER COMMENTS

Reviewer #3 (Remarks to the Author):

The authors have satisfactorily answered my question about the ClyA-luciferase construct and its surface exposure.

However, my second point about OMV composition, especially with regards to LPS, remains. The authors now state that the LPS content of the OMVs is 120 ng/ml with an OMV protein concentration of 1 mg/ml, so about 0.01 %. This is extremely low, OMVs isolated without detergent extraction typically have LPS concentrations in the range 10-20%, and 1-2% after detergent treatment. This makes me wonder how the LPS concentration was determined, but no details are given. It reinforces my concern about proper OMV characterization. Immunization results critically depend on OMV properties and composition, and in the present manuscript these are not described in sufficient detail. However, this is essential for interpretation and comparison of the results with other studies in the field.

Point-by-point responses to the reviewer

Note: Following are our responses (in blue color) to reviewers' comments (in bold black color) and sentences described in the revised manuscript are highlighted in yellow.

Reviewer 3

The authors have satisfactorily answered my question about the ClyA-luciferase construct and its surface exposure.

However, my second point about OMV composition, especially with regards to LPS, remains. The authors now state that the LPS content of the OMVs is 120 ng/ml with an OMV protein concentration of 1 mg/ml, so about 0.01 %. This is extremely low, OMVs isolated without detergent extraction typically have LPS concentrations in the range 10-20%, and 1-2% after detergent treatment. This makes me wonder how the LPS concentration was determined, but no details are given. It reinforces my concern about proper OMV characterization. Immunization results critically depend on OMV properties and composition, and in the present manuscript these are not described in sufficient detail. However, this is essential for interpretation and comparison of the results with other studies in the field.

Response: We are grateful for the reviewer to point out the importance of LPS in the vaccine formulation. Regarding the contents of LPS in OMVs, we have thoroughly consulted the literatures for the quantitative measurement of LPS in OMVs. According to the testing methods, the reported LPS contents of OMVs have significantly different values for different sources of OMVs by different methods¹⁻⁴.

Waterbeemd et al. utilized a modified gas chromatography method to quantify LPS content of OMVs, and the LPS content of OMVs was about 243 µg/mg OMVs (24.3%, the mass ratio of LPS to OMVs total protein)¹, which is consistent with the description of the reviewer. In the gas chromatography method, LPS was isolated by hot phenol-water extraction and quantified using the peak height of C14:0-3OH with C12:0-2OH as the internal standard (two C14:0-3OH residues per LPS).

Compared to the gas chromatography method, the limulus amoebocyte lysate (LAL) assay is a functional assay and can reflect the activity of LPS. It uses a colorimetric method in which endotoxin catalyzes the activation of a proenzyme in LAL, which

will cleave a colorless substrate to produce a colored end-product. LAL assay has been widely used as the gold standard in pharmaceutical, clinical and scientific fields for the detection of bacterial endotoxin. Pfalzgraff et al. quantified LPS content of OMVs by LAL assay, and the LPS content of OMVs was 13 µg/mg (1.3%)². In another study³, the LPS content of OMVs was reported as 4 EU/mg, which is equal to 400 pg/mg (0.00004%)³, according to the equation, 100 pg = 1 EU assumed to convert endotoxin mass to activity². Vanaja et al. reported that the LPS content of OMVs was 1.3 µg/mg OMVs (0.13%) quantified by LAL assay⁴. Therefore, although using the same LPS detection method, the reported LPS contents of OMVs still have significantly different values on a huge range.

Using the LAL assay to analyze the LPS content of OMVs, our result (204.1 ng/mg, 0.02%) is closed to that in the reference 4 (0.13%). We then carefully compared the methods used in our study and the literature. We found that, compared to the OMVs extraction method used in the reference 4, we applied an additional ultrafiltration concentration step using a 50-K ultrafiltration tube and a filtration step using 0.22 µm filter membrane to concentrate the isolated OMVs. When we used the same OMVs extraction method without ultrafiltration and filtration, the LPS content of the OMVs was about 596.6 ng/mg (0.06%, which is three times higher than our original measurement). Given the high sensitivity of LAL detection and the difference between bacterial strain, these differences of LPS contents are within the acceptable range.

In addition, to eliminate interference from other substances in the endotoxin, we used ELISA to specifically detect the LPS in OMVs. The results were 49.7 ng/mg OMVs (about 0.005%) and lower than the result measured by LAL assay (0.02%).

We summarized the LPS detection methods using both ELISA and LAL assay in the SI as follows.

LAL assay kit is designed as a quantitative assay that is simple and sensitive for detection of the presence of LPS in the samples. It uses a colorimetric method in which endotoxin catalyzes the activation of a proenzyme in LAL, which will cleave a colorless substrate to produce a colored end-product. The end-product can be measured spectrophotometrically and compared to a standard curve. In this experiment, we used a known concentration of LPS to convert a unit of EU/mL into ng/mL.

ELISA assay employs the competitive inhibition enzyme immunoassay technique. A

monoclonal antibody specific to LPS has been pre-coated onto a microplate. A competitive inhibition reaction is launched between biotin labeled LPS and unlabeled LPS (standards or samples) with the pre-coated antibody specific to LPS. After incubation, the unbound conjugate is washed off. Next, avidin conjugated to horseradish peroxidase (HRP) is added to each microplate well. The amount of bound HRP conjugate is reversely proportional to the concentration of LPS in the samples. After addition of the substrate solution, the intensity of color developed is reversely proportional to the concentration of LPS in the sample.

The LPS related results were added into the manuscript as follows.

The lipopolysaccharides (LPS) content in CC OMV was 49.9 and 204.1 ng/mg OMV protein measured by enzyme-linked immunosorbent assay (ELISA) and limulus amoebocyte lysate (LAL) assay, respectively. (Page 8, Line 10)

The LPS content in OMV was detected by ELISA (CEB526Ge, Cloud-Clone Corp., Wuhan, China) and LAL assay (L00350C, GenScript, Nanjing, China).

References

1. Waterbeemd, B. V. D., Streefland, M., Ley, P. V. D., et al. Improved OMV vaccine against *Neisseria meningitidis* using genetically engineered strains and a detergent-free purification process. *Vaccine* **28**, 4810-4816 (2010).
2. Pfalzgraff, A., Correa, W., et al. LPS-neutralizing peptides reduce outer membrane vesicle-induced inflammatory responses. *BBA-MOL CELL BIOL L* **1864**, 1503-1513 (2019).
3. Ahmadi, B. S., Moshiri, A., Ettehad, M. F., et al. Extraction and Evaluation of Outer Membrane Vesicles from Two Important Gut Microbiota Members, *Bacteroides fragilis* and *Bacteroides thetaiotaomicron*. *Cell J.* **22**, 344-349 (2020).
4. Sivapriya, K. V., Ashley, J., et al. Bacterial Outer Membrane Vesicles Mediate Cytosolic Localization of LPS and Caspase-11 Activation. *Cell* **165**, 1106-1119 (2016).

REVIEWERS' COMMENTS

Reviewer #3 (Remarks to the Author):

In the revision the authors have provided more information on OMV characterization, especially with regards to details of the method for LPS content determination.